# A purely Quantum Generative Modeling through Unitary Scrambling and Collapse

## Abstract

Quantum computing offers fundamentally more expressive mechanisms for generative modeling, yet current approaches remain constrained by classical neural components that bottleneck quantum capability and hardware efficiency. We propose the Quantum Scrambling and Collapse Generative Model (QGen), a purely quantum paradigm that eliminates classical architectural dependencies. QGen implements two coherent processes: scrambling, which interleaves Gaussian diffusion channels with unitary delocalization to disperse information globally while avoiding collapse into uninformative states; and collapse, where parameterized quantum circuits refocus scrambled distributions into structured outputs, achieving distributional reconstruction under coherent evolution. To enable scalability, we introduce a measurement-based training principle that decomposes learning into tractable subproblems, mitigating barren plateaus. Empirically, QGen outperforms classical and hybrid baselines under matched parameter budget, while maintaining robustness under finite-shot sampling, demonstrating strong feasibility for near-term hardware.

## 1 Introduction

Generative modeling has emerged a central theme in modern machine learning, with paradigms such as Generative Adversarial Networks (GANs) (Goodfellow et al., 2014a), Variational Autoencoders (VAEs) (Kingma & Welling, 2014), and Denoising Diffusion Probabilistic Models (DDPMs) (Ho et al., 2020a) driving progress in data generation. Despite these successes, classical models typically require large neural architectures, costly training, and extensive parameters to achieve state-of-the-art performance. Quantum computing offers a principled alternative: by leveraging superposition and entanglement, quantum systems can represent and manipulate complex probability distributions with intrinsic parallelism (Nielsen & Chuang, 2010), providing compact yet expressive models whose representational capacity scales exponentially with qubit count. These features point to a compelling pathway toward quantum-native generative modeling (Preskill, 2018; Lloyd & Weedbrook, 2018; Benedetti et al., 2019).

While promising, these initial efforts largely represent pragmatic compromises rather than a paradigmatic shift. They predominantly employ quantum circuits as subroutines within classically prescribed frameworks. For instance, QGANs employ quantum generators but rely on classical discriminators (Situ et al., 2020), quantum Boltzmann machines (QBMs) (Amin et al., 2018) and QVAEs integrate quantum-structured latent spaces while relying on classical optimization (Khoshaman et al., 2018), and quantum diffusion models such as QuDDPM (Kölle et al., 2024) depend on classical backbones (e.g., U-Nets) to approximate gradients. This persistent reliance inherits classical bottlenecks and limits expressivity in the Hilbert space.

Moreover, when deployed on real quantum hardware, hybrid models face additional inefficiencies: treating quantum and classical processors as disjoint instruments with separate data transfer pipelines forces repeated context switching and intermediate data exchange. The resulting latency prevents classical decisions from influencing the quantum state before qubits decohere, further limiting the feasibility of hybrid schemes(Lubinski et al., 2022). Consequently, current approaches fall short of enabling a purely quantum learning process and restrict the potential for genuine quantum advantage in generative modeling (Huang et al., 2021).

In this work, we propose the QGen, a purely quantum framework for image synthesis. QGen eliminates all classical architectural dependencies by grounding the generative process in two coherent mechanisms: scrambling and collapse. The scrambling operator interleaves Gaussian diffusion channels with unitary delocalization, dispersing local perturbations into global quantum correlations (Hosur et al., 2016). The Gaussian component regularizes the trajectory, preventing convergence to maximally mixed states and yielding a tractable Gaussian prior, while the unitary component preserves coherence and enriches expressivity through interference and entanglement. The collapse operator, implemented via parameterized quantum circuits (PQCs)(Cerezo et al., 2021a), then inverts only the delocalization dynamics, as the Gaussian component is analytically tractable. This suffices to refocus scrambled information into structured distributions, establishing a coherent and quantum-native reversal mechanism.

Training QGen departs from conventional likelihood-based approaches. Instead of relying on intractable objectives, we introduce a quantum-native training strategy that leverages measurement to define tractable learning signals, enabling effective optimization while preserving scalability on near-term hardware. Empirically, QGen validates this paradigm: it *outperforms classical and hybrid baselines under comparable parameter budgets*. Moreover, it *maintains stability* under finite-shot conditions and robustness to statistical noise, underscoring both the theoretical sufficiency and the practical feasibility of the scrambling–collapse framework for near-term quantum hardware.

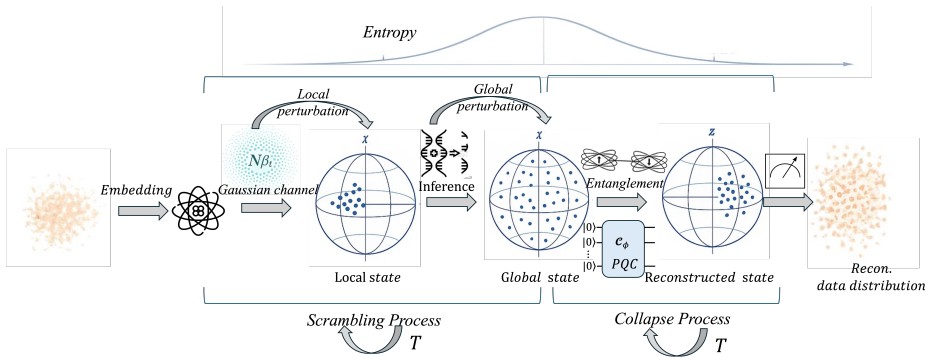

Figure 1: Schematic of the proposed quantum generative model. Classical data are embedded into quantum states and progressively scrambled by local and global perturbations. The resulting global state is then collapsed by a parameterized quantum circuit to reconstruct the data distribution.

## 2 RELATED WORK

**Fully Quantum Generative Models.** Recent work in quantum machine learning distinguishes hybrid and fully quantum approaches to generative modeling, with the latter seeking to exploit the expressive capacity of quantum states absent classical dependencies (Cerezo et al., 2022). Quantum Born Machines represent data distributions through measurement probabilities of parameterized quantum states and are trained via likelihood-based or moment-matching criteria (Liu & Wang, 2018; Benedetti et al., 2019). Quantum Boltzmann and related energy-based models define target distributions via effective Hamiltonians and Gibbs states (Kieferová & Wiebe, 2017). Quantum diffusion proposals such as QuDDPM investigate stochastic quantum channels and unitary scrambling to synthesize distributions over quantum states (Zhang et al., 2024). While these methods advance fully quantum perspectives, empirical evaluations have largely focused on low-dimensional or domain-specific settings such as synthetic quantum states or physics data (Zoufal et al., 2019; Gao & Duan, 2017), and often require classical latent encoding (e.g., PCA, Autoencoders) to interface with high-dimensional images (Khoshaman et al., 2018; Chang et al., 2024; Stein et al., 2021). As a result, a scalable, end-to-end quantum pathway for image synthesis remains underexplored.

**Algorithmic and hardware limitation.** The chasm between theoretical promise and practical realizability in quantum machine learning is exacerbated by fundamental constraints of near-term variational quantum algorithms. Gradient concentration phenomena and barren plateaus impede optimization scalability with increasing circuit width or depth, particularly for unstructured ansatzes (McClean et al., 2018; Cerezo et al., 2021b; Cerezo de la Roca et al., 2021). Quantum data encod-

ing and access impose overheads that often dominate end-to-end computational complexity (Schuld et al., 2021; Rath & Date, 2024). Furthermore, finite-shot statistical estimation, decoherence effects (Kandala et al., 2019), and repetitive quantum-classical communication bottlenecks degrade training fidelity and throughput on existing hardware. These limitations collectively motivate algorithmic designs that preserve native quantum information pathways, regularize stochastic quantum dynamics to prevent entropy homogenization, and decompose learning objectives into computationally tractable subproblems while minimizing quantum circuit depth and classical post-processing overhead.

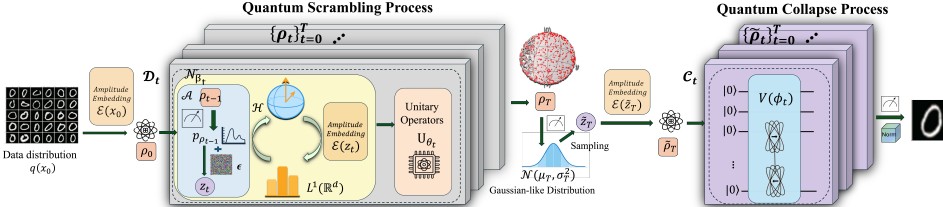

Figure 2: The Purely Quantum Scrambling-Collapse Generative Framework. (a) Forward Scrambling Process (left): an initial data sample $x_0 \sim q(x_0)$ progressively transformed by alternating Gaussian diffusion channels $\mathcal{N}_{\beta_t}$ and unitary delocalization operators $U_{\theta_t}$, dispersing local information into global quantum correlations and yielding a scrambled state $\rho_T$ whose measurement statistics converge to a Gaussian-like prior; (b) Reverse Collapse Process (right): Parameterized collapse operators $\{\mathcal{C}_t\}_{t=1}^{T}$ (detailed in Appendix A.2) are trained to iteratively refocus the scrambled information, reversing the dynamics to reconstruct the target data distribution from the simple prior.

## 3 METHOD

The overall framework is summarized in Figure 2. Formally, $x_0$ denotes a random variable sampled from an unknown data distribution $q(x_0)$ with access only to samples rather than the density itself. The model constructs a quantum dynamical process $\{\rho_t\}_{t=0}^{T}$ by preparing an initial state $\rho_0 = \mathcal{E}(x_0)$ via amplitude encoding. This state is then evolved under a scrambling operator $\mathcal{D}$, designed to delocalize quantum information, yielding a terminal state $\rho_T = \mathcal{D}(\rho_0)$ that is highly scrambled and exhibits measurement statistics approaching a tractable prior distribution (e.g., Gaussian distribution). Formalizing this generative modeling task as a thermodynamic recovery problem, the objective of QGen is to learn the inverse mapping along this scrambling path, enabling the recovery of structured data distributions by transforming $\rho_T$ back to $\rho_0$.

### 3.1 SCRAMBLING AS DELOCALIZATION

The scrambling operator $\mathcal{D}$ progressively transforms an initial quantum state $\rho_0$ into a delocalized state $\rho_T$ through two fundamental components: i) a Gaussian diffusion channel $\mathcal{N}_{\beta_t}$ introducing controlled stochasticity, and ii) a unitary delocalization operator $U_{\theta_t}$, that globally redistributes these perturbations across the Hilbert space. Formally, the process is a composition of $T$ steps:

$$\rho_T = \left( \mathcal{D}_T \circ \cdots \circ \mathcal{D}_1 \right)(\rho_0), \quad \text{where} \quad \mathcal{D}_t(\cdot) = U_{\theta_t} \mathcal{N}_{\beta_t}(\cdot) U_{\theta_t}^{\dagger}. \tag{1}$$

The condition for unitary inversion is formalized via thermodynamic recoverability, quantified by the Coherent Information $I_c$ Schumacher & Nielsen (1996). More formally,

**Definition 3.1.** *Let $R$ be a reference system purifying the input state $\rho_{t-1}$, and $Q_t$ denote the output system containing the scrambled state $\rho_t = \mathcal{D}_t(\rho_{t-1})$. The coherent information of the state $\rho_t$ is defined as:*

$$I_c(R \rangle Q_t)_{\rho_t} \equiv S(\rho_t) - S(\rho_{R,t}), \tag{2}$$

*where $S(\rho_{R,t})$ represents the joint entropy of $R$ and $Q_t$. While classical distribution recovery allows weaker bounds (e.g., Holevo information (Holevo, 1973)), our protocol relies on unitary inversion, for which $I_c$ serves as the strict reversibility criterion (Barnum et al., 1998).*

**Theorem 3.2.** *For the collapse map $\mathcal{C}_t$ to recover the target state $\rho_{t-1}$ via unitary inversion, the scrambling process requires the preservation of the Coherent Information: $I_c(R\rangle Q_t)_{\rho_t} \geq S(\rho_{t-1})$.*

This recoverability condition follows from the coherent-information bound of (Schumacher & Nielsen, 1996). The detailed proof context is given in Appendix C.1.

**Gaussian diffusion channel $\mathcal{N}_{\beta_t}$.** This channel introduces controlled stochasticity by perturbing the classical probability distribution associated with $\rho_{t-1}$. The procedure prevents native quantum noise (e.g., depolarizing channels) or excessive scrambling from driving the system toward the maximally mixed state $\rho = I/d$, which would erase useful structure and render the generative process uninformative (Deutsch, 1991; Gogolin & Eisert, 2016; Plenio & Vitelli, 2001). Instead, Gaussian perturbations are selected to maximize thermodynamic diversity while preserving a smooth, learnable geometry. This ensures convergence to a tractable prior (statistically validated in Appendix F.1), enabling a well-defined reversal mapping. The channel operation proceeds in two stages:

$$\mathcal{N}_{\beta_t}(\rho_{t-1}) = (\mathcal{E} \circ \mathcal{A})(\rho_{t-1}) = |\psi_t\rangle\langle\psi_t|. \tag{3}$$

First, a noise injection map $\mathcal{A} : \mathcal{D}(\mathcal{H}) \rightarrow L^1(\mathbb{R}^d)$ extracts the classical probability vector $p_{t-1}$ from $\rho_{t-1}$ via projective measurement and perturbs it with Gaussian noise:

$$\mathbf{z}_t = \sqrt{\bar{\alpha}_t}\, p_{t-1} + \sqrt{1 - \bar{\alpha}_t}\, \boldsymbol{\epsilon}, \quad \boldsymbol{\epsilon} \sim \mathcal{N}(0, I), \tag{4}$$

where $\bar{\alpha}_t = \prod_{s=1}^{t}(1 - \lambda_g \beta_s)$ for a predefined variance schedule $\{\beta_t\}_{t=1}^{T}$. The hyperparameter $\lambda_g \in [0, 1]$ regulates the noise intensity, balancing entropy growth against recoverability.

Subsequently, an amplitude embedding map $\mathcal{E} : L^1(\mathbb{R}^d) \rightarrow \mathcal{D}(\mathcal{H})$ encodes the noised distribution into a quantum state:

$$\mathcal{E}(\mathbf{z}_t) = |\psi_t\rangle\langle\psi_t|, \quad \text{where} \quad |\psi_t\rangle = \sum_{i=1}^{d} \sqrt{\mathbf{z}_t^i}\, |i\rangle\,. \tag{5}$$

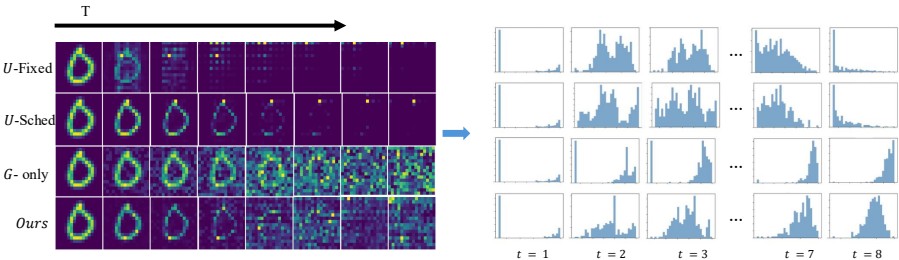

Figure 3: Visualization of scrambling strategies. Left: Comparison of noise scheduling strategies: U-Fixed (purely unitary), U-Sched (unitary with variance scheduling), G-only (Gaussian channel only), and Ours. Pure unitary dynamics induce progressive entropy homogenization, driving the system toward the maximally mixed state. Introducing variance scheduling or Gaussian noise regulates entropy growth and preserves recoverable structure. Right: Log-scaled measurement distributions across timesteps $t$, showing that the joint action of noise injection and unitary scrambling drives smooth convergence toward a Gaussian-like prior, thereby ensuring statistical tractability and reversible dynamics.

**Unitary delocalization operator $U_{\theta_t}$** This operator globally redistributes local perturbations into complex correlations across the Hilbert space. At each step t, the angles $\theta_t$ are independently sampled from a normal distribution $\mathcal{N}(0, \sigma_t^2 I)$, where the variance $\sigma_t^2 = 1 - \bar{\alpha}_t$ is modulated by the noise schedule. This coordination ensures that the scrambling intensity increases coherently with the injected noise level, preventing the premature loss of structural information through overly aggressive perturbation in early steps. The resulting operation is defined as:

$$\rho_t = U_{\theta_t}(\mathcal{N}_{\beta_t}(\rho_{t-1})) = U_{\theta_t}\, \mathcal{N}_{\beta_t}(\rho_{t-1})\, U_{\theta_t}^{\dagger}, \tag{6}$$

where $U_{\theta_t}$ is a unitary circuit composed of local rotations that are alternately applied to different qubits. This evolution enhances expressivity beyond the Gaussian-like marginal statistics through: (i) constructive and destructive interference of the injected noise patterns during global redistribution, which generates a richer set of perturbation modes beyond classical noise (Benedetti et al., 2021); and (ii) the introduction of randomly parameterized unitary dynamics, which leverage coherent scrambling to promote exploration of a larger volume of the Hilbert space (Anschuetz & Kiani, 2022) and helps mitigate mode collapse in the generative learning process.

As illustrated in Figure 3, the synergy between $\mathcal{N}_{\beta_t}$ and $U_{\theta_t}$ induces a coherent scrambling trajectory. The diffusion channel contributes controlled stochasticity but, when used alone, produces local and uncorrelated perturbations with low recoverability. The unitary delocalization step transforms these perturbations into global correlations, organizing the evolution into a correlation-rich pathway in Hilbert space. This coordinated mechanism disperses information without collapsing into uninformative fixed points and ensures that the forward trajectory remains amenable to reliable inversion via PQCs, a property that is essential for scalability on near-term devices.

## 3.2 Collapse as Reconstruction

The scrambling process yields a terminal state $\rho_T$ whose measurement statistics empirically converge to a Gaussian-like distribution. This distribution, induced by the stochastic scrambling dynamics, can be approximated in practice by estimating its mean $\mu_T$ and variance $\sigma_T^2$ from measurement outcomes. The resulting Gaussian prior serves as a tractable initialization point for the reverse phase.

Instead of reconstructing directly from a single scrambled state, we sample a classical vector $\tilde{\mathbf{z}}_T \sim \mathcal{N}(\mu_T, \sigma_T^2)$ from this induced prior, and prepare its quantum embedding $\tilde{\rho}_T = \mathcal{E}(\tilde{\mathbf{z}}_T)$ as the starting state of the collapse process.

The reconstruction is performed by a parameterized collapse operator $\mathcal{C}_\phi$, which maps $\tilde{\rho}_T$ back to a quantum state $\tilde{\rho}_0$ whose measurement statistics align with the target data distribution $q(x_0)$. Formally, the reverse process is defined as the composition of learned quantum steps:

$$\tilde{\rho}_0 = \mathcal{C}_\phi(\tilde{\rho}_T) = (\mathcal{C}_1 \circ \mathcal{C}_2 \circ \cdots \circ \mathcal{C}_T)(\tilde{\rho}_T), \tag{7}$$

where each $\mathcal{C}_t$ is an independently trained PQC implementing a unitary transformation $V(\phi_t)$. To reflect the growing complexity of inverting deeper scrambling, the expressive capacity of $\mathcal{C}_t$ increases with decreasing $t$, thereby allocating greater representational power to later steps. This ensures that residual perturbations are progressively corrected (Nichol & Dhariwal, 2021) and structural information is restored in a stable manner, preparing the system for alignment with the original data distribution.

**Theoretical Sufficiency of Unitary Reverse Steps.** The efficacy of a unitary reverse process $\mathcal{C}_t$ stems from the structure of the scrambling step $\mathcal{D}_t$: Gaussian noise is applied classically with known statistics, while subsequent scrambling is unitary and information-preserving (Wilde, 2013). The role of $V_{\theta_t}$ is thus not to reverse noise injection directly, but to invert its delocalization—concentrating dispersed information back into structured distributions. Specifically, $V_{\theta_t}$ is trained to transform scrambled states into outputs whose measurement statistics match the less noisy states $\rho_{t-1}$. This is feasible due to the universal approximation capability of parameterized quantum circuits (Schuld et al., 2021; Mitarai et al., 2018), which can approximate such statistical inversions via coherent evolution, effectively refocusing scrambled information into the target distribution.

The quantum-native approach offers inherent advantages over classical neural networks: (1) it preserves quantum coherence throughout the entire generative process, enabling efficient representation of complex data correlations through superposition and entanglement (Benenti et al., 2001); (2) it avoids the representational bottleneck of classical networks by leveraging the high-dimensional state space, whose expressivity grows exponentially with qubit count (Abbas et al., 2021); and (3) it ensures native compatibility with the forward quantum process, eliminating the need for costly quantum-classical data conversions and enabling end-to-end quantum generative learning (Lubinski et al., 2022).

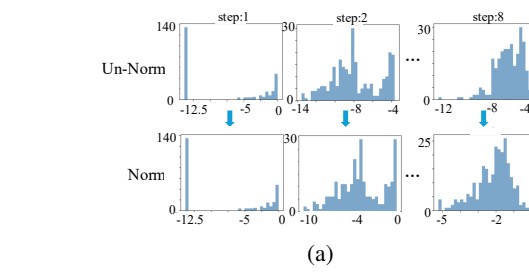 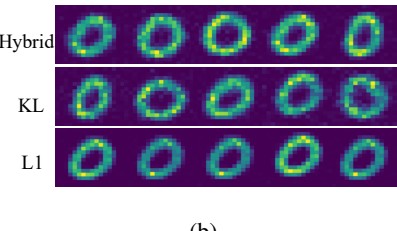

(a)                                                                    (b)

Figure 4: Effect of normalization and loss design. (a) Log-scaled probability distributions of quantum states across timesteps with and without normalization. Normalization compensates for exponentially small probabilities, often spanning several orders of magnitude, thereby alleviating sparsity issues induced by the exponentially large state space, preserving the relative entropy structure required for divergence minimization. (b) Reconstructed samples under different training losses, highlighting the trade-off between diversity and fidelity.

## 3.3 TRAINING OBJECTIVE AND OPTIMIZATION

Our training strategy is conceptually inspired by the evidence lower bound (ELBO) in classical diffusion models, which reformulates log-likelihood maximization as a sequence of distribution-matching objectives across timesteps (Ho et al., 2020b; Sohl-Dickstein et al., 2015). In the quantum setting, however, the non-Gaussian and unitary nature of scrambling dynamics precludes closed-form derivations, and the states $\rho_t$ admit no efficient classical representation (Gao & Duan, 2017), rendering direct ELBO maximization intractable.

To address this, we introduce a measurement-based surrogate objective that preserves the inductive bias of stepwise distribution alignment while remaining computable on quantum hardware. Specifically, each collapse operator $\mathcal{C}_t$ is trained to minimize the divergence between the reconstructed state $\tilde{\rho}_{t-1} = \mathcal{C}_t(\tilde{\rho}_t)$ and the target $\rho_{t-1}$, where $\tilde{\rho}_t$ is produced by the previously trained steps $\{\mathcal{C}_s\}_{s=t+1}^{T}$ applied to $\tilde{\mathbf{z}}_T$. Since the scrambling process is ultimately assessed via measurement, this distribution-matching objective provides a necessary and sufficient criterion for inversion. The composite loss functional is:

$$\mathcal{L}(\phi_t) = \mathbb{E}_{x_0 \sim q(x_0)} \left[ \lambda_{\mathrm{KL}} \, D_{\mathrm{KL}} \left( \tilde{p}_{\rho_{t-1}} \, \| \, \tilde{p}_{\tilde{\rho}_{t-1}} \right) + \lambda_{\mathrm{L1}} \left\| \tilde{p}_{\rho_{t-1}} - \tilde{p}_{\tilde{\rho}_{t-1}} \right\|_1 \right], \tag{8}$$

where $p_\rho(x) = \langle x | \rho | x \rangle$ denotes the computational-basis measurement distribution. To render this objective computationally tractable, we employ renormalized distributions:

$$\tilde{p}_\rho(x) = \frac{p_\rho(x)}{\max_{x' \in \{0,1\}^n} p_\rho(x')}, \tag{9}$$

which compensate for exponentially small probabilities and improve numerical stability. Equation 8 balances two principles: the KL term enforces global alignment and maintains diversity, while the $L_1$ term provides local accuracy and preserves fine-grained structure. As shown in Figure 4, normalization stabilizes histograms and the combined loss balances diversity and fidelity.

More fundamentally, the measurement-based objective induces a geometric structure that relaxes the optimization landscape compared to full state tomography.

**Theorem 3.3.** *The distributional loss $\mathcal{L}(\phi_t)$ expands the target from a unique unitary to a Phase-Invariant Solution Manifold: $\mathcal{M} = \{\mathcal{C}_\phi : diag(\mathcal{C}_\phi(\tilde{\rho}_t)) = diag(\rho_{t-1})\}$. This manifold invariance removes stiff phase-dependent modes, significantly improving the effective condition number of the Quantum Fisher Information Matrix (QFIM) (Cerezo et al., 2021b).*

Nonetheless, learnability holds only within a restricted regime, determined jointly by the geometric structure (Theorem 3.3) and the thermodynamic recoverability bound (Theorem 3.2).

**Corollary 3.4** (The Fisher-Coherent Critical Window)**.** *The efficient trainability of the ansatz $\mathcal{C}_\phi$ is constrained to a scrambling window $[t_{scr}, t_{BP}]$. The lower bound ($t > t_{scr}$) necessitates sufficient operator spreading to ensure that the Fisher Information Matrix is full-rank and coherent information remains recoverable. Conversely, the upper bound ($t < t_{BP}$) requires the trajectory to*

*remain pre-thermal; as scrambling approaches a unitary 2-design, the gradient variance vanishes as* $\text{Var}(\nabla\mathcal{L}) \sim \mathcal{O}(2^{-N})$*, rendering the model untrainable regardless of the manifold geometry.*

Consequently, optimal generative performance exists uniquely in the intermediate window where the system is sufficiently chaotic to generate complex correlations, yet remains thermodynamically reversible and geometrically well-conditioned. Empirically, we observe a coherent information value of $\bar{I}_c \approx 1.5$ nats at the optimal depth, consistent with Theorem 3.2. Further details on the empirical validation and underlying mechanisms of this critical window are provided in Appendices C.3.

**Contrast with Classical Diffusion.** Unlike classical diffusion models that rely on dissipative stochasticity ($dS > 0$), QGen exploits the deterministic unitary reversibility of quantum chaos ($\Delta S_{\text{global}} = 0$).QGen harnesses this by targeting the phase-invariant manifold $\mathcal{M}$ (Theorem 3.3) within the pre-thermal window (Corollary 3.4), replacing probabilistic score estimation with unitary trajectory unwinding.

The complete training and sampling procedures are formalized in Algorithms 1 and 2, respectively. Optimization proceeds per timestep, with gradients estimated from quantum measurements, effectively decomposing the inverse problem into tractable subproblems and mitigating the barren-plateau effect that afflicts end-to-end training of deep quantum circuits (see Appendix D for gradient-variance and convergence analyses).

---

**Algorithm 1** Quantum Scrambling-Collapse Training

**Require:** Data distribution $q(x_0)$, schedule $\{\beta_t\}_{t=1}^T$
**Require:** Collapse circuits $\{\mathcal{C}_t\}_{t=1}^T$
1: **for** $t = T$ to $1$ **do**
2:    **repeat**
3:       $x_0 \sim q(x_0)$
4:       $\rho_t \leftarrow (\mathcal{D}_t \circ \cdots \circ \mathcal{D}_1)(\mathcal{E}(x_0))$
5:       $\tilde{\mathbf{z}}_T \sim \mathcal{N}(\mu_T, \sigma_T^2)$
6:       $\tilde{\rho}_t \leftarrow (\mathcal{C}_{t+1} \circ \cdots \circ \mathcal{C}_T)(\mathcal{E}(\tilde{\mathbf{z}}_T))$
7:       Measure $p_{\rho_t}, p_{\tilde{\rho}_t}$
8:       $\tilde{p}_{\rho_t} \leftarrow p_{\rho_t} / \max p_{\rho_t}$; $\tilde{p}_{\tilde{\rho}_t} \leftarrow p_{\tilde{\rho}_t} / \max p_{\tilde{\rho}_t}$
9:       $\mathcal{L}(\phi_t) \leftarrow \lambda_{\text{KL}} D_{\text{KL}}(\tilde{p}_{\rho_t} \| \tilde{p}_{\tilde{\rho}_t}) + \lambda_{\text{L1}} \|\tilde{p}_{\rho_t} - \tilde{p}_{\tilde{\rho}_t}\|_1$
10:    **until** Convergence
11: **end for**

**Algorithm 2** Quantum Sampling

**Require:** Trained collapse circuits $\{\mathcal{C}_t\}_{t=1}^T$
**Require:** Prior $\mu_T, \sigma_T^2$, samples $n$
1: $P \leftarrow \emptyset$
2: **while** $|P| < n$ **do**
3:    $\tilde{\mathbf{z}}_T \sim \mathcal{N}(\mu_T, \sigma_T^2)$
4:    $\tilde{\rho}_T \leftarrow \mathcal{E}(\tilde{\mathbf{z}}_T)$
5:    **for** $t = T$ to $1$ **do**
6:       $\tilde{\rho}_{t-1} \leftarrow \mathcal{C}_t(\tilde{\rho}_t)$
7:    **end for**
8:    $x_0 \leftarrow \tilde{p}_{\tilde{\rho}_0}(x) = \langle x|\tilde{\rho}_0|x\rangle$
9:    $P \leftarrow P \cup \{x_0\}$
10: **end while**
11: **return** $P$

---

## 4 EXPERIMENTS

We evaluate the proposed QGen framework on three image benchmarks of increasing complexity: MNIST, Fashion MNIST, and an EMNIST subset (letters a–c) at 16×16 resolution. This setup allows us to assess the model's ability to generalize from simple patterns to more structured data while remaining compatible with current quantum hardware limits. Performance is reported using Fréchet Inception Distance (FID), computed per class with 5k samples. Additional results on full resolution (28×28) and multi class generation are provided in Appendix F.4.

We compare QGen against three representative baselines: a state of the art classical diffusion model (DDPM with a U Net backbone), standard classical generators (DCGAN, VAE), and hybrid quantum classical models (QVUNet, PQWGAN). This comparison situates QGen within the broader generative modeling landscape and highlights the benefits of a fully quantum generative pathway.

### 4.1 COMPARISON TO THE BASELINE

Table 1 demonstrates the compelling performance of the proposed QGen framework. Under comparable parameter budgets ($N_\theta \approx 3\text{k} \sim 6\text{k}$), QGen achieves state-of-the-art FID scores on MNIST (168.54) and Fashion-MNIST (214.76), while maintaining competitive performance on the more challenging EMNIST dataset (307.68).

A key finding is the consistent outperformance of QGen over hybrid quantum-classical baselines, as demonstrated by its superior FID scores across all datasets (e.g., 168.54 vs. 282.65 on MNIST).

Table 1

| Method Category | Method | $N_\theta$ | MNIST | FashionMNIST | EMNIST |
|---|---|---|---|---|---|
| Classical | GAN(Goodfellow et al., 2014b) | 3456 | 203.67 | 252.95 | 315.31 |
| | Diffusion(De Falco et al., 2024) | 4661 | 186.32 | 234.51 | **296.41** |
| | Normalization Flow (Dinh et al., 2016) | 3232 | 240.00 | 275.06 | 311.84 |
| | VAE(Kingma & Welling, 2013) | 3249 | 195.44 | 233.25 | 309.67 |
| Hybrid | QVUNet(De Falco et al., 2024) | 5917 | 282.65 | 245.09 | 324.20 |
| | PQWGAN(Tsang et al., 2023) | 4426 | 242.85 | 305.91 | 358.49 |
| Purely Quantum | QuantumGen | 3090 | **168.54** | **214.76** | 307.68 |

This performance gap reveals a fundamental limitation of hybrid paradigms: their iterative quantum-classical feedback introduces decoherence through repetitive measurements, disrupting essential quantum information processing. In contrast, QGen's purely quantum architecture maintains coherent evolution throughout the generative process.

This coherent pathway enables more effective harnessing of quantum resources. The framework leverages quantum interference to create richer perturbation modes during scrambling, surpassing classical noise injection. Simultaneously, maintained entanglement efficiently encodes complex pixel correlations that would require larger parameter budgets in classical networks.

These results position QGen not merely as a competitor to classical models, but as a evidence for a fundamentally more efficient scaling paradigm. The ability to achieve superior generative fidelity with constrained resources highlights the potential of quantum-native approaches to overcome the parametric bottlenecks of classical deep learning.

## 4.2 ABLATION STUDIES

We perform ablation studies on the MNIST (digits 0) subset to examine the necessity of each design choice in QGen. Specifically, we analyze scrambling components, variance allocation, and model capacity to disentangle their roles in shaping the statistical trajectory and in enabling effective reconstruction. In each ablation study, different variants of QGen models are trained with the setting in Sec. 4.1.

Table 2: Ablation study on quantum scrambling components.

| Components | | | $T = 6$ | $T = 7$ | $T = 8$ | $T = 9$ | $T = 10$ |
|---|---|---|---|---|---|---|---|
| $\{U_{\theta_t}\}_{t=1}^T$ | $\{\mathcal{N}_{\beta_t}\}_{t=1}^T$ | $\{\sigma_t^2\}_{t=1}^T$ | FID($\downarrow$) | FID($\downarrow$) | FID($\downarrow$) | FID($\downarrow$) | FID($\downarrow$) |
| ✓ | × | × | 315.02 | 321.56 | 283.37 | 347.64 | 362.74 |
| ✓ | × | ✓ | 301.41 | 314.98 | 291.47 | 290.93 | 301.45 |
| × | ✓ | × | 276.94 | 257.23 | 202.19 | 224.33 | 213.16 |
| ✓ | ✓ | ✓ | **206.44** | **193.26** | **161.76** | **163.02** | **179.83** |

**Ablation on scrambling components.** Table 2 disentangles the contributions of unitary delocalization $\{U_{\theta_t}\}_{t=1}^T$, Gaussian diffusion channels $\{\mathcal{N}_{\beta_t}\}_{t=1}^T$, and variance scheduling $\{\sigma_t^2\}_{t=1}^T$. Purely unitary dynamics yield the weakest results, with FID worsening as $T$ grows, reflecting entropy homogenization that progressively erodes recoverable structure. Adding variance scheduling moderates this effect by distributing scrambling intensity more smoothly, but the absence of stochastic convergence still limits reversibility. Gaussian diffusion alone achieves better FID by driving convergence, yet performance saturates with larger $T$ as the model lacks quantum-aligned expressivity. The complete configuration consistently delivers the best results (FID 162–206), visual comparison in Figure 5 demonstrates the clear superiority of coordinated noise and delocalization, generating sharp and diverse samples that closely resemble the target distribution.

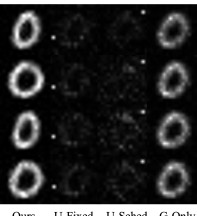

Ours  U-Fixed  U-Sched  G-Only

Figure 5: Comparison of generated samples under different scrambling strategies.

Notably, high-fidelity generation is achieved with only $T \leq 10$ steps representing $a \sim 100\times$ reduction compared to classical diffusion ($T > 1000$). This efficiency stems from the synergy between Gaussian channels and unitary scrambling, which balances entropy growth with expressive coherence. Such efficiency offers a promising approach to overcome the depth and sampling constraints of near-term quantum hardware, demonstrating that scrambling–collapse models can attain practical generative performance under realistic resource budgets.

**Ablation on variance allocation mechanisms.** Table 3a and 3b jointly evaluate two complementary factors that govern variance allocation: the scheduling of $\{\beta_t\}_{t=1}^{T}$ across timesteps, and the global intensity regulator $\lambda_g$. The cosine schedule consistently achieves the lowest FID, indicating that smooth variance progression avoids both early under-regularization and late-stage entropy saturation, thereby maintaining reversibility. Similarly, moderate values of $\lambda_g$ yield superior performance by balancing entropy growth against recoverability, whereas excessively large values induce over-homogenization of measurement statistics and degrade inversion. These results highlight the critical role of our designed variance allocation in stabilizing scrambling trajectories and ensuring tractable reconstruction.

Table 3: Results of ablation studies. (a) Comparison of variance schedules $\{\beta_t\}_{t=1}^{T}$; (b) Comparison of the quantum perturbation intensity regulator $\lambda_g$; (c) Comparison of parameterized collapse operator $\mathcal{C}_\phi$ depth and model capacity.

| (a) | | | (b) | | | (c) | | |
|---|---|---|---|---|---|---|---|---|
| $\{\beta_t\}_{t=1}^{T}$ | FID($\downarrow$) | | $\lambda_g$ | FID($\downarrow$) | | $l_0$ | $N_\theta$ | FID($\downarrow$) |
| Cosine | **161.76** | | 0.1 | **161.76** | | 6 | 1650 | 219.47 |
| Linear | 203.67 | | 0.3 | 180.11 | | 8 | 2130 | 190.87 |
| Sigmoid | 240.67 | | 0.5 | 190.65 | | 10 | 2610 | 165.89 |
| Log | 232.56 | | 0.7 | 185.28 | | 12 | 3090 | **161.76** |
| | | | 0.9 | 215.41 | | 14 | 3570 | 242.27 |

**Ablation on model capacity.** As shown in Table 3c, varying depth from 6 to 14 layers reveals a fundamental quantum-classical divergence: PQC performance non-monotonically depends on complexity due to the barren plateau phenomenon, where gradient variances scale inversely with Hilbert space dimension. Our hierarchical training strategy mitigates this by decomposing the learning task into locally optimizable subproblems, constraining the optimization landscape to maintain measurable gradients. Results demonstrate that while sufficient depth enables complex quantum transformations, excessive layers induce both coherent noise accumulation and gradient evaporation, degrading generalization performance.

### 4.3 ROBUSTNESS TO STATISTICAL NOISE

Practical quantum hardware operates with finite sampling that introduces statistical noise, unlike the ideal infinite-shot assumption in previous evaluations. This section quantitatively assesses QGen's robustness under such realistic constraints.

**Experimental Setup.** Let $\rho_\infty$ denote the quantum state prepared under the ideal infinite-shot measurement regime, with corresponding measurement distribution $p_\infty(x) = \langle x|\rho_\infty|x\rangle$ and generated image $\mathcal{I}_\infty$. To quantify the robustness to finite-shot noise, we fix the pretrained model parameters $\theta^*$ and generate quantum states $\rho_N$ through projective measurement with $N = 2^k$ shots for $k = 5, \ldots, 14$.

Figure 6 presents a visual analysis of the shot-convergence behavior for both MNIST and FashionMNIST datasets. The generated images exhibit rapid perceptual convergence, with visual fidelity approaching the infinite-shot baseline $\mathcal{I}_\infty$ at $N_{\text{shots}} \geq 2048$.

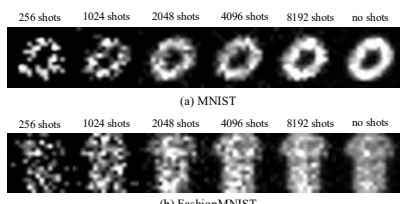

Figure 6: Generated images with varying shots for (a) MNIST and (b) FashionMNIST.

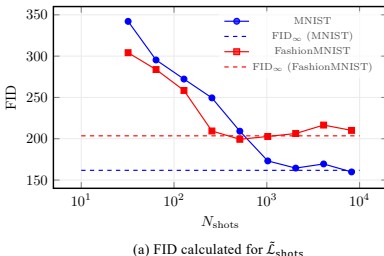 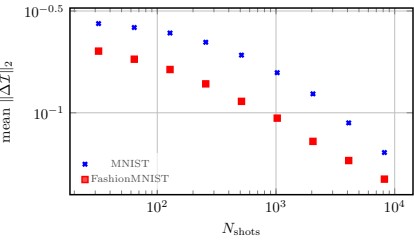

(a) FID calculated for $\tilde{\mathcal{L}}_{\text{shots}}$        (b) Pixel-by-pixel $L2$ distance between $\tilde{\mathcal{L}}_{\text{shots}}$ and $\tilde{\mathcal{L}}_{\infty}$

Figure 7: Image generation quality with different measurement shots. (a) FID values computed for the infinite-shot reference $\mathcal{I}_{\infty}$ across varying shot counts. (b) Pixel-wise $L_2$ distance between finite-shot approximations $\mathcal{I}_{\text{shots}}$ and the infinite-shot baseline $\mathcal{I}_{\infty}$ on MNIST dataset.

This visual convergence is quantitatively validated in Figure 7. While pixel-wise $L_2$ distance (Figure 7b) decreases exponentially with shot count as expected from statistical sampling theory ($\sim \mathcal{O}(1/\sqrt{N_{shots}})$) (Nielsen & Chuang, 2010), the perceptual FID metric (Figure 7a) converges rapidly to near-baseline values at $N_{\text{shots}} \geq 2048$. This divergence indicates that numerical pixel-level deviations become perceptually insignificant well before mathematical convergence, affirming the practical viability of finite-shot sampling.

These findings confirm that QGen remains effective even under low-shot conditions. The model's robustness to statistical noise not only enhances its practicality for near-term quantum hardware, but also supports the use of finite sampling budgets without compromising image fidelity.

## 5    Scalability Analysis and Limitation

We analyze the QGen's scalability through the lens of optimization geometry and hardware feasibility. Algorithmically, the stepwise decomposition acts as an architectural regularizer that maintains non-vanishing gradients; as detailed in Appendix D, gradient variance decays polynomially with circuit depth, mitigating the barren plateau problem inherent to deep global circuits. From a hardware perspective, the framework leverages sampling-based access to avoid full tomography; empirical analysis in Appendix E indicates that sample complexity scales polynomially, governed by the effective dimension of the data manifold. The generative performance further exhibits a stability plateau under realistic physical noise levels ($p \leq 10^{-2}$), suggesting that the non-local encoding of information provides robustness against local NISQ errors.

Current implementations, however, remain subject to NISQ-era constraints. The reliance on exact amplitude encoding constitutes an $\mathcal{O}(2^n)$ initialization bottleneck, which, compounded by limited qubit resources, currently restricts experiments to simple grayscale image datasets. However, QGen's modular design decouples the generative backbone from data loading. Consequently, scaling to complex domains will necessitate concurrent advances in qubit resources and efficient ansätze design, alongside the integration of approximate state-preparation schemes.

## 6    Conclusion

This work introduces QGen, a quantum generative modeling framework grounded in the principle of scrambling and collapse. By replacing likelihood-based objectives and latency-prone hybrid schemes with a self-contained quantum process, we demonstrate that effective quantum generation need not rely on computationally intractable training signals. Theoretically, QGen integrates stochastic regularization with unitary delocalization into a coherent and principled alternative. Empirically, it surpasses classical and hybrid baselines under comparable parameter constraints and remains robust under both finite-shot execution and moderate NISQ noise levels, confirming its practical viability on near-term devices. This framework establishes a scalable and inherently quantum-native path toward generative modeling, marking a concrete step toward practical quantum advantage as hardware continues to mature.

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

# A    QUANTUM COMPUTING BACKGROUND

## A.1    BASIC CONCEPTS

**Quantum Computing.** Quantum computing leverages the fundamental principles of quantum mechanics, namely superposition, entanglement, and unitary evolution, to process information in ways that can surpass classical computational limits for certain problems.

**Qubits and Superposition.** The basic unit of quantum information is the *qubit*, a two-level quantum system with computational basis states $|0\rangle$ and $|1\rangle$. A general pure state can be expressed as

$$|\psi\rangle = \alpha|0\rangle + \beta|1\rangle, \quad \text{with } |\alpha|^2 + |\beta|^2 = 1, \tag{10}$$

where $\alpha, \beta \in \mathbb{C}$ are complex probability amplitudes. When both $\alpha$ and $\beta$ are non-zero, the qubit is in a superposition of the basis states.

**Quantum Entanglement.** Entanglement is a form of correlation unique to quantum mechanics. A multi-qubit state such as the Bell state

$$|\Phi^+\rangle = \frac{1}{\sqrt{2}}(|00\rangle + |11\rangle) \tag{11}$$

cannot be factored into a product of single-qubit states. Entanglement is widely regarded as a key enabler of quantum advantage in machine learning and generative tasks, as it allows non-classical correlations to be encoded compactly (Horodecki et al., 2009).

**Quantum Measurement.** Measurement in quantum mechanics is described by positive operator-valued measures (POVMs). For a state $\rho$ measured in the computational basis $\{|i\rangle\}$, the probability of outcome $i$ is given by the Born rule:

$$p(i) = \text{Tr}(|i\rangle\langle i|\rho) = \langle i|\rho|i\rangle. \tag{12}$$

The measurement process projects the quantum state onto the observed basis state, providing the fundamental mechanism for extracting classical data from quantum generative models.

**Unitary Evolution.** The evolution of a closed quantum system is described by unitary operators $U$, ensuring that the state transformation $|\psi'\rangle = U|\psi\rangle$ preserves normalization. Parameterized quantum circuits (PQCs) implement such unitaries with tunable gate parameters, enabling trainable quantum feature maps (Cerezo et al., 2021a; Schuld et al., 2020). In the context of diffusion and flow-based models, PQCs serve as expressive transformations capable of scrambling and denoising data distributions.

**Quantum Gate Set.** Fundamental quantum gates implement unitary transformations on qubit states. The Pauli gates $\{X, Y, Z\}$ provide basis rotations, while the Hadamard gate $H$ creates superposition states from computational basis states. Two-qubit gates such as the controlled-NOT (CNOT) gate generate entanglement, forming the foundation for universal quantum computation. In parameterized quantum circuits (PQCs), gates like $R_X(\theta), R_Y(\theta), R_Z(\theta)$ with tunable parameters $\theta$ enable expressive quantum transformations for machine learning tasks.

**Density Operators and Mixed States.** While pure states provide an intuitive description, realistic systems are often described by a density operator $\rho$, which encodes both pure and mixed states:

$$\rho = \sum_i p_i |\psi_i\rangle\langle\psi_i|, \quad p_i \geq 0, \sum_i p_i = 1. \tag{13}$$

Mixed states naturally capture uncertainty and decoherence, and are particularly relevant when analyzing measurement-induced dynamics or noise-injected quantum diffusion.

**Hilbert Space Dimension and Expressivity.** An $n$-qubit system resides in a Hilbert space of dimension $d = 2^n$. This exponential scaling enables compact representation of complex high-dimensional distributions, offering potential advantages in generative modeling compared to classical approaches (Gao et al., 2022). However, it also makes classical simulation intractable, underscoring the need for quantum-native algorithms.

**Scrambling and Thermalization.** A concept increasingly relevant in quantum generative modeling is *scrambling*, which describes how local quantum information becomes delocalized across many

degrees of freedom (Hosur et al., 2016). Scrambling underpins the design of quantum diffusion processes, where initially structured data are evolved into highly mixed states approximating simple priors (e.g., uniform or Gaussian-like distributions). The reverse process can then be learned to reconstruct structured data from scrambled quantum states.

## A.2 PARAMETERIZED QUANTUM CIRCUIT

PQCs are quantum circuits containing gates with tunable parameters $\theta$. They implement a parameterized unitary transformation $U(\theta)$ that prepares a quantum state $|\psi(\theta)\rangle$ from an initial state. The parameters $\theta$ are optimized classically to minimize a cost function, enabling the circuit to learn specific tasks. In generative modeling, PQCs serve as the trainable quantum channel that evolves a prior distribution into a target data distribution.

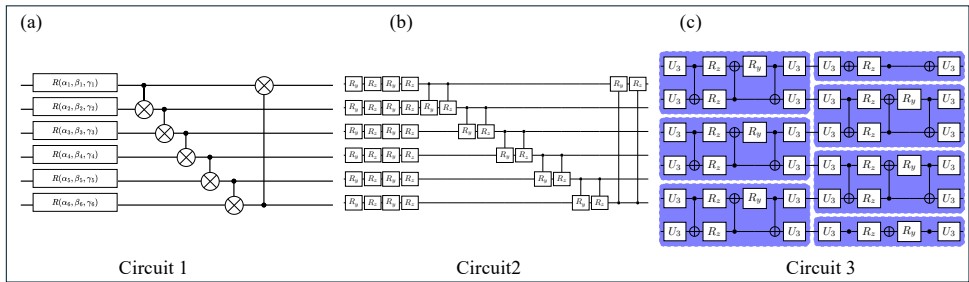

Figure 8: PQCs architecture used as collapse operator $\mathcal{C}_\phi$. (a) Circuit 1 is based on PennyLane's `qml.StronglyEntanglingLayers` (a dense circuit), inspired by circuit-centric quantum classifiers Schuld et al. (2020). (b) Circuit 2 is from a quantum GAN papers for continuous data generation Bravo-Prieto et al. (2022). (c) Circuit3 is composed of repeated two-qubit quantum circuits (blue square), responsible for an arbitrary $SU(4)$ state generation MacCormack et al. (2022).

Figure 8 and Table 4 present an ablation study on the expressivity-capacity trade-off in parameterized quantum circuits employed as the collapse operator $\mathcal{C}_\phi$ during the reverse generative process. The empirical evidence reveals a non-monotonic relationship between circuit expressivity and model performance. Specifically, while the parameter count $N_\theta$ increases substantially across the architectural hierarchy (Circuit 1 $\rightarrow$ 3), the corresponding FID exhibits degradation on the MNIST dataset.

Table 4: Ablation study on the modeling capacity of PQCs. The FID is evaluated for 12-layer circuits with increasing parameter counts $N_\theta$ on MNIST (16×16), averaged over all digit classes (0-9) using 5,000 generated samples. Circuit 1 achieves the best performance despite having fewer parameters, indicating that larger PQCs do not necessarily improve generation quality for this task.

| Circuit | $N_\theta$ | FID($\downarrow$) |
|---------|------------|--------------------|
| Cir.1 | 3090 | **168.54** |
| Cir.2 | 6180 | 190.44 |
| Cir.3 | 15459 | 195.89 |

This phenomenon can be understood through the lens of quantum optimization landscapes. Excessive parameterization in PQCs is known to induce barren plateaus, where the gradient variance vanishes exponentially with system size, thereby impeding convergence. Moreover, the increased Hilbert space exploration capacity of larger circuits may lead to overfitting to noise patterns rather than learning the underlying data manifold.

Theoretical work by Cerezo et al. (2021a) suggests that there exists an optimal circuit depth and parameter count for a given task complexity, beyond which the signal-to-noise ratio in gradient estimation deteriorates. Our results align with this hypothesis, indicating that for MNIST ($8 \times 8$ resolution), the task complexity is insufficient to warrant the full expressivity of Circuit 3.

However, as demonstrated in Appendix F for higher-dimensional data ($28 \times 28$ images), the increased representational capacity of Circuit 3 becomes advantageous. This bifurcated behavior underscores a fundamental principle in quantum machine learning: the optimal architectural complexity is intrinsically tied to the spectral properties and dimensionality of the target data distribution. The transition point where additional quantum resources yield diminishing returns thus serves as an important hyperparameter in quantum generative modeling.

## B  EXPERIMENT DETAILS

### B.1  QGEN

**Model configuration.** The scrambling process is executed for $T = 8$ steps with a variance schedule $\{\beta_t\}_{t=1}^T$ following a cosine profile, modulated by a global intensity regulator $\lambda_g = 0.1$. For the reverse generative process, Circuit 1 serves as the collapse operator $\mathcal{C}_\phi$, implemented using $n = 8$ data qubits and $n_A = 2$ auxiliary qubits. The circuit depth is initialized at 12 and increases linearly with $t$, enabling progressively more expressive transformations during the reverse process. The training objective combines a KL divergence term and an $\ell_1$-based fidelity term, weighted by $\lambda_{\text{KL}} = 0.5$ and $\lambda_{\ell_1} = 5$, respectively, to balance sample diversity with reconstruction fidelity.

**Training Details.** All experiments on the MNIST, FashionMNIST, and EMNIST datasets use images resized to $16 \times 16$. For each class, 1,000 training samples are provided, with a batch size of 32. The proposed QGen framework is implemented in PennyLane, leveraging its PyTorch interface to enable end-to-end classical backpropagation. Optimization is performed with Adam at a learning rate of $10^{-3}$ for 80 epochs, coupled with a StepLR scheduler with step size 10 and decay factor $\gamma = 0.05$ to improve convergence stability.

### B.2  CLASSICAL MODELS

**DDPM Baseline.** We implement a U-Net–based DDPM with base dimension $d = 4$ and multipliers $\text{dim\_mults} = (1, 2)$, forming a hierarchical encoder–decoder architecture. The baseline incorporates ResNet blocks with group normalization (groups=2), sinusoidal time embeddings, symmetric skip connections, and transposed convolutions for upsampling.

The model is trained using a cosine variance schedule over $T = 1,000$ diffusion steps, with a $\ell_2$ noise-prediction objective to balance gradient contributions across noise levels. Optimization is performed with the Adam optimizer (learning rate $10^{-3}$, $\beta_1 = 0.9$, $\beta_2 = 0.99$) enhanced by exponential moving averaging (EMA) for stable convergence. All models are trained for 9,375 steps with a batch size of 32 using 1,000 samples per class.

**GAN Baseline.** We implement a Deep Convolutional Generative Adversarial Network (DCGAN) with symmetric generator and discriminator architectures. The generator processes a 5-dimensional latent vector through sequential transposed convolutional layers with feature map dimensions of 16 and 8, while the discriminator employs strided convolutional layers with feature maps expanding from 4 to 8, using LeakyReLU activations (negative slope=0.2) for stable adversarial training.

The model is trained with binary cross-entropy loss (BCELoss) where real and fake labels are set to 1 and 0, respectively. Both generator and discriminator are optimized using Adam with learning rate 0.0002 and $\beta_1$=0.5. Training runs for 200 epochs with batch size 12, using fixed noise batches for progression monitoring.

**VAE Baseline.** We implement a Vanilla Variational Autoencoder (VAE) with a 2-dimensional latent space. The encoder consists of two convolutional layers with feature dimensions [4, 16], each employing strided convolutions (kernel size=3, stride=2), batch normalization, and LeakyReLU activations. The resulting features are mapped to mean ($\mu$) and log-variance ($log \, \sigma^2$) vectors through linear projections.

The decoder mirrors this architecture in reverse, employing transposed convolutions for upsampling and `Tanh` activations, and reconstructs the input through a final convolutional layer. The model is trained with a composite loss that combines reconstruction error and KL divergence, weighted by a factor of 0.00025. Optimization is performed using Adam (learning rate = 0.005) with exponential learning rate scheduling ($\gamma = 0.95$) over 40 epochs with a batch size of 128.

**NF Baseline.** We implement a RealNVP architecture with a multi-scale coupling structure, configured with 2 scales and 8 coupling blocks per transformation. The model processes three-channel input images (`in_channels` = 3) with a 0.9 data constraint for numerical stability before applying invertible transformations. Each scale employs alternating checkerboard and channel-wise masking patterns in affine coupling layers with 64 intermediate channels. The model is trained for 100 epochs with batch size 128 using Adam optimization (learning rate = 0.001), gradient clipping (max norm = 100), and L2 regularization (weight decay = $5 \times 10^{-5}$) applied specifically to weight norm scale factors.

### B.3 HYBRID MODELS

**QVUNet Baseline.** We implement a quantum-enhanced U-Net architecture (QVUNet) that integrates quantum convolutional layers within a classical diffusion framework. The model employs a base dimension of 4 with multiplicative factors $(1, 2)$ across scales. The core component is the Quantum Residual Block, which replaces standard convolutional operations with parameterized quantum circuits.

Each quantum block processes input features through 13 parallel quantum channels using the HQ-Conv ansatz with 3 layers (28 parameters per layer). The quantum circuit operates on 8-qubit systems, processing local $2 \times 2$ image patches through variational quantum circuits. The hybrid architecture concatenates quantum-processed features with classical features, followed by group normalization. The model shares identical training configurations with the classical U-Net baseline, including the cosine variance schedule, Adam optimization, and equivalent hyperparameters.

**PQWGAN Baseline.** We implement a Patch Quantum Wasserstein GAN (PQWGAN), a hybrid quantum–classical model with a quantum generator and a classical discriminator. The quantum generator consists of 11 layers, each applied to 8 qubits without ancilla qubits. The circuit encodes latent vectors via parameterized rotations followed by entangling CNOT layers, producing probability distributions that are post-processed into image patches. Since we use a single generator with patch size 16×16, the model outputs full grayscale images of dimension 16×16.

The discriminator is fully classical, comprising two lightweight fully connected layers. The first layer maps the flattened image input to 16 hidden units with a LeakyReLU activation (negative slope = 0.2), followed by a final linear layer that outputs a scalar Wasserstein score. Training is performed using the Wasserstein loss with gradient penalty, a batch size of 128, and optimization over 50 epochs.

## C THEORETICAL ANALYSIS

### C.1 PROOF CONTEXT OF THEOREM 3.2

**Proof Context.** Let $\rho_{t-1}$ be the input state and $\rho_t = \mathcal{D}_t(\rho_{t-1})$ the scrambled output. The inequality $I_c(R\rangle Q_t)_{\rho_t} \geq S(\rho_{t-1})$ is sufficient for full quantum state recovery; since matching Born probabilities is strictly weaker than reconstructing the full density matrix, this condition is more than adequate for distributional recovery.

For our mechanism of *unitary unwinding*, however, satisfying this bound is also necessary. Although weaker information-theoretic criteria may suffice for recovering classical statistics, approximating the inverse unitary propagator requires that the coherent information remain non-negative. When the system over-scrambles (e.g., $t \to \infty$) such that $I_c < 0$, the forward channel becomes *antidegradable* (Barnum et al., 1998), meaning that the environment holds stronger correlations with the reference $R$ than the system $Q_t$ does. In this regime, the quantum capacity vanishes, and the pre-scrambled information cannot be recovered by any operation acting solely on $Q_t$. Thus, maintaining $I_c \geq S(\rho_{t-1})$ ensures that the scrambling trajectory remains within a reversible regime accessible to the ansatz.

### C.2 PHYSICAL MECHANISMS OF THE CRITICAL WINDOW

While Corollary 3.4 defines the operational bounds $[t_{\text{scr}}, t_{\text{BP}}]$, the underlying physical mechanisms driving this phase transition warrant further elaboration:

**The Under-Scrambling Regime** ($t \ll t_{\mathrm{scr}}$): In this shallow depth limit, the model failure is not due to trainability, but triviality. As shown in Larocca et al. (2023), the Quantum Fisher Information Matrix (QFIM) suffers from rank deficiency, meaning the ansatz lacks the expressivity to generate the complex correlations required to match the target distribution.

**The Over-Scrambling Regime** ($t \gg t_{\mathrm{BP}}$): Conversely, at deep depths, the failure mechanism shifts to untrainability. The system approaches a unitary 2-design, causing the optimization landscape to flatten exponentially (Barren Plateaus). Furthermore, in the presence of any non-unitarity, coherent information would decay, rendering the process information-theoretically irreversible.

### C.3    EMPIRICAL VALIDATION

Theorem 3.2 posits that successful generation requires $I_c \geq S(\rho_{t-1})$. We validated this requirement by computing the exact average von Neumann entropies for all $N = 5,920$ training trajectories.

We observe a sustained coherent information of $\bar{I}_c \approx 1.5$ nats at the optimal training depth ($T = 8$). Given that the entropy of the target distribution is approximately $S(\rho_T) \approx 1.5$ nats, this observation confirms that:

$$I_c \approx S(P_{\mathrm{data}}). \tag{14}$$

This equality implies that the scrambling process is effectively *lossless* with respect to the generative potential. The channel capacity remains strictly above the "Black Hole" limit ($I_c \to 0$), indicating that the backward ansatz is tasked with unwinding a deterministic unitary trajectory rather than reversing entropic thermalization.

Furthermore, this recoverable regime is finite. We observe that the subsequent collapse of training performance at deeper circuits coincides precisely with the predicted transition to the untrainable phase described in Corollary 3.4, serving as a negative control that validates the upper bound of our critical window.

## D    OPTIMIZATION STABILITY AND GRADIENT ANALYSIS

### D.1    GRADIENT VARIANCE ANALYSIS

We analyze the optimization landscape of QGen by examining the scaling of the gradient variance, $\mathrm{Var}(\partial_{\theta_k}\mathcal{L})$, relative to circuit depth $L$ and system size $n$. Prior work shows that sufficiently deep, unstructured PQCs approximate unitary 2-designs, under which gradients obey concentration-of-measure and decay exponentially as

$$\mathrm{Var}(\partial_{\theta_k}\mathcal{L}) \in \mathcal{O}(e^{-\alpha n}), \tag{15}$$

for some $\alpha > 0$ (McClean et al., 2018). This exponential suppression gives rise to the barren plateau phenomenon and renders gradient-based optimization intractable for high-dimensional Hilbert spaces.

The QGen framework circumvents this pathological scaling through a time-sliced decomposition strategy. Instead of learning a global unitary transformation $\mathcal{U}_{global}$ with a single deep circuit ($L \in \Omega(n)$ depending on exactness), we approximate the generative trajectory via a composition of $T \in \Omega(n/\log(n))$ sequential operators $\{\mathcal{C}_t\}_{t=1}^T$. This distributes the representational burden such that each operator operates within a logarithmic-depth regime, typically

$$L = \mathcal{O}(\log n). \tag{16}$$

In this depth range, local-cost analyses predict that gradient variance decays at most polynomially,

$$\mathrm{Var}(\partial_{\theta_k}\mathcal{L}) \in \Omega(\mathrm{poly}(n^{-1})), \tag{17}$$

thus preserving trainability in practice (Cerezo et al., 2021b).

Empirical analysis, as presented in Figure 9, corroborates this theoretical expectation. Across depths $L \in [4, 100]$ and qubit counts $n \in \{6, 8, 10\}$, the variance remains within $\mathcal{O}(10^{-2}) - \mathcal{O}(10^{-3})$ for moderate depths and exhibits polynomial, rather than exponential, decay with system size. A sharp transition to the barren plateau regime is only observed at excessive depths ($L \approx 100$). Since the

QGen architecture employs shallow sub-circuits (experimentally configured at $L \leq 20$ per step), the model is strictly confined within the trainable phase of the optimization landscape.

These results jointly validate that the stepwise collapse mechanism functions as an architectural regularizer that prevents gradient concentration, ensuring stable and non-vanishing gradient variance throughout optimization on near-term devices.

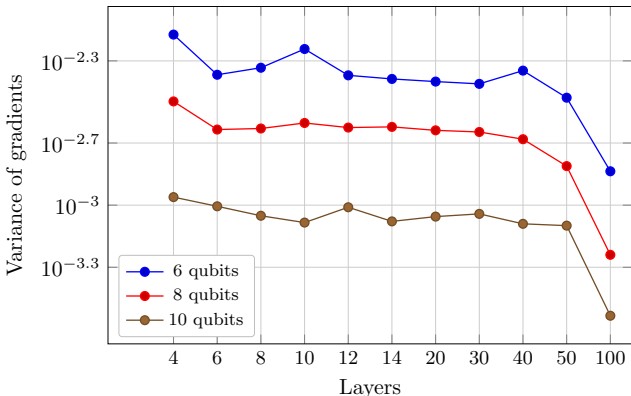

Figure 9: Gradient variance scaling analysis. The variance showing polynomial decay ($\mathcal{O}(10^{-2}) - \mathcal{O}(10^{-3})$)) up to for depths $L \leq 50$, confirming that QGen's shallow sub-circuits remain well within the trainable regime. A sharp transition to vanishing gradients is only observed at excessive depths ($L = 100$).

### D.2 CONVERGENCE BEHAVIOR

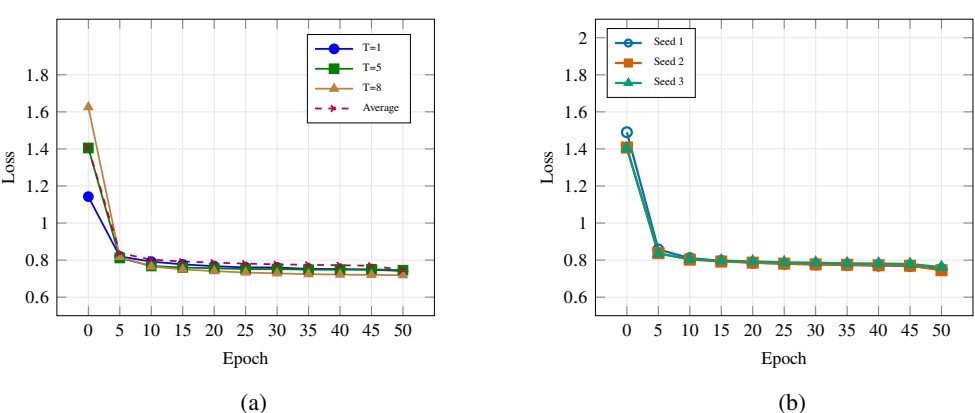

(a)  (b)

Figure 10: Convergence behavior of QGen: (a) loss trajectories during training for selected timesteps $t \in \{1, 5, 8\}$; (b) stability of the averaged loss $L_{\text{avg}}$ across three random initializations.

To complement the gradient–variance study, we further analyze the dynamical behavior of the stepwise objective during training. Figure 10 reveals several structural properties of the optimization landscape. Deeper scrambling depths $T$ lead to larger initial losses, reflecting the increased complexity of inverting highly entangled states and the larger Hilbert-space distance between the initialized ansatz and the target distribution. Nevertheless, all trajectories converge smoothly toward a stable region with nearly identical rates. This indicates that the measurement-based objective sustains a consistent gradient flow toward the minimizer, and crucially, that the problem conditioning does not deteriorate significantly as the scrambling depth $T$ increases.

Moreover, the averaged-loss trajectories from three independent random initializations are almost perfectly aligned, suggesting that the optimization dynamics are governed by a broad basin of attraction within the explored initialization regime. Such structural stability implies that the landscape

explored by QGen resides in an effectively convex regime, in which the time-sliced decomposition acts as an implicit regularizer that suppresses the pathological curvature variations typically found in deep variational circuits. The resulting convergence behavior is therefore robust and reproducible across different random initializations.

Together with the gradient-variance study, these observations provide theoretical and empirical support that the stepwise training principle induces a well-conditioned optimization geometry, enabling stable learning of the effective reverse transformation on near-term quantum devices.

# E ROBUSTNESS AND SAMPLING ANALYSIS

## E.1 QUANTUM NOISE ROBUSTNESS

Table 5: Robustness against Physical Noise. FID scores under Depolarizing and Amplitude Damping channels with varying error probabilities $p$. A broad stability region persists for $p \leq 10^{-2}$, indicating that the generative trajectory remains well-behaved under realistic perturbation strengths.

| Noise Model | Sampling | Noise Strength ($p$) | | | | | | | |
|---|---|---|---|---|---|---|---|---|---|
| | | 0 (Base) | 1e-4 | 5e-4 | 1e-3 | 5e-3 | 1e-2 | 5e-2 | 1e-1 |
| Depolarizing | Ideal ($\infty$) | 160.98 | 184.04 | 183.53 | 185.17 | 184.39 | 180.78 | 191.22 | 208.76 |
| | Finite (2k) | 160.98 | 194.29 | 204.13 | 200.31 | 205.77 | 216.29 | 250.29 | 279.89 |
| Amp. Damping | Ideal ($\infty$) | 160.98 | 180.45 | 183.35 | 183.02 | 182.69 | 178.80 | 185.46 | 242.31 |
| | Finite (2k) | 160.98 | 199.03 | 209.30 | 200.96 | 210.24 | 214.03 | 258.07 | 284.82 |

To evaluate the generative stability of QGen under realistic hardware imperfections, we inject depolarizing and amplitude-damping noise at the level of single-qubit gates during the collapse process. These two noise models probe complementary non-ideal behaviors: depolarizing noise introduces uniform incoherent fluctuations, whereas amplitude damping drives relaxation toward the ground state. The noise strengths $p \in [10^{-4}, 10^{-1}])$ span the operative range of small-scale superconducting and trapped-ion devices.

Table 5 reports FID under two evaluation modes: analytic (no-shots) density-matrix simulation and a 2048-shot sampling regime. The transition from $p = 0$ to $p > 0$ marks a fundamental shift from the pure-state manifold to the convex set of mixed states, explaining the discrete FID shift observed at the first noise tier ($p = 10^{-4}$). Beyond this structural transition, however, the system exhibits a broad stability plateau for $p \leq 10^{-2}$. In this regime, performance degrades smoothly and remains well-behaved, indicating that the learned reverse transformations deform continuously under small noise perturbations. This behavior suggests that the information encoded by the scrambling process is sufficiently non-local and redundant such that local gate-level noise injected during collapse training does not destroy the global structure required for reconstruction. As a result, the inverse mapping learned by the collapse circuits remains stable under noise levels typical of current NISQ hardware.

Comparison between analytic and sampled settings further shows that finite-shot sampling contributes only additive deviations and does not amplify noise-induced perturbations. No instability, divergence, or abrupt collapse is observed across the studied range. These results demonstrate that QGen remains robust under representative NISQ noise conditions. While scaling to larger systems will ultimately require improvements in hardware fidelity, the forward–reverse decomposition retains structural stability under standard noise profiles and supports reliable training at present device scales.

## E.2 MEASUREMENT SHOT SCALING

In Section 4.3, we showed that QGen matches the infinite-shot baseline once the sampling budget reaches roughly 2048 shots. However, to rigorously verify whether this measurement requirement scales exponentially with the number of qubits, a potential bottleneck for larger systems, we extended our evaluation to variable system sizes. We systematically measured generative fidelity across qubit counts $n \in \{9, 10, 11, 12\}$ under increasing shot budgets ranging from 256 to 8192.

Table 6: Measurement Scaling Analysis. FID scores across increasing qubit counts and shot budgets. The "convergence point" where finite-shot performance matches the ideal baseline remains consistent at $\approx 1024$–$2048$ shots regardless of system size.

| $n$ | No-shots | 256 | 512 | 1024 | 2048 | 4096 | 8192 |
|---|---|---|---|---|---|---|---|
| 9 | 183.04 | 205.62 | 192.75 | 182.20 | 183.02 | 186.72 | 184.27 |
| 10 | 160.98 | 249.46 | 209.16 | 173.16 | 164.47 | 169.49 | 159.83 |
| 11 | 196.34 | 255.51 | 226.74 | 206.44 | 209.49 | 208.54 | 209.62 |
| 12 | 234.92 | 276.49 | 251.49 | 229.70 | 234.13 | 230.17 | 246.98 |

To isolate the effect of system size, we maintained a fixed data resolution of $16 \times 16$ (requiring 8 data qubits) and increased the total Hilbert space dimension by introducing auxiliary qubits ($n_{aux} = n - 8$)

As shown in Table 6, generative fidelity improves substantially as shots increase from 256 to 1024 but stabilizes beyond 2048 shots across all system sizes ($n = 9$–$12$). For instance, with 12 qubits ($d = 4096$), increasing the budget from 2048 to 8192 shots yields negligible improvement (FID $234.13 \rightarrow 246.98$). Crucially, this convergence threshold remains constant as $n$ increases, suggesting that sample complexity is governed by the effective dimension of the learned data manifold rather than the full Hilbert space dimension ($2^n$). This suggests that, within the qubit ranges tested, the number of required measurements is governed by sampling accuracy rather than an exponential dependence on system size.

# F    MORE EXPERIMENT RESULTS

## F.1    GAUSSIAN-LIKE MEASUREMENT STATISTICS

To empirically assess whether the marginal measurement statistics of QGen approach a Gaussian-like form (as visually suggested in Figure 3), we evaluated the terminal scrambling distributions using standard normality tests. Across 5,920 histogram instances, both the Shapiro–Wilk and D'Agostino–Pearson tests yielded $p$-values that frequently exceeded the $\alpha = 0.05$ threshold, indicating that the null hypothesis of normality could not be rejected for the majority of samples. These results provide supportive evidence that the repeated Gaussian perturbations in the forward process induce marginal distributions that are well-approximated by a Gaussian within the tested regime.

## F.2    ABLATION ON SCRAMBLING INTENSITY

## F.3    QUALITATIVE COMPARISON WITH BASELINES

## F.4    SCALABILITY AND MULTI-CLASS GENERATION

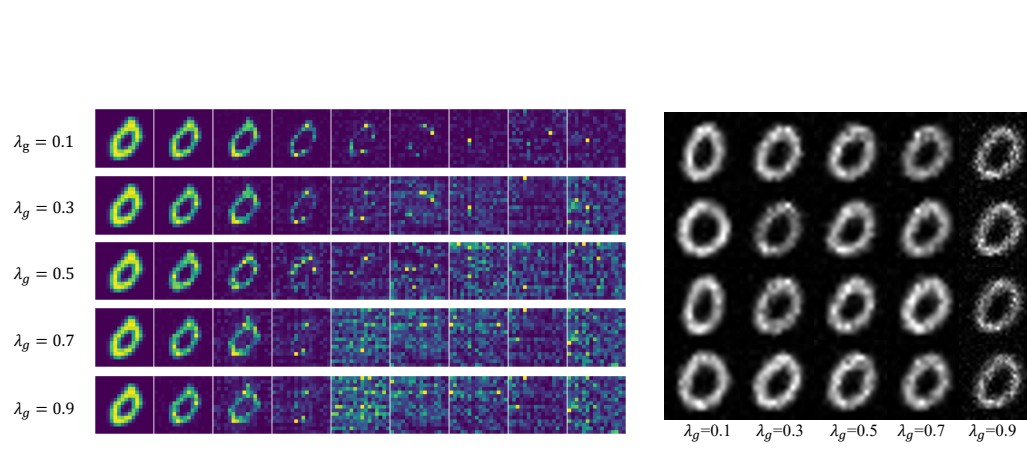

$\lambda_{\text{g}} = 0.1$
$\lambda_g = 0.3$
$\lambda_g = 0.5$
$\lambda_g = 0.7$
$\lambda_g = 0.9$

$\lambda_g$=0.1    $\lambda_g$=0.3    $\lambda_g$=0.5    $\lambda_g$=0.7    $\lambda_g$=0.9

Figure 11: Visualization of the effect of the global intensity regulator $\lambda_g$. Left: intermediate states under different $\lambda_g$ values during the scrambling process. Right: corresponding generated MNIST samples. Moderate values of $\lambda_g$ (e.g., 0.3–0.5) yield superior performance by balancing entropy growth with recoverability. In contrast, larger $\lambda_g$ accelerates information loss due to stronger perturbations, leading to degraded sample quality.

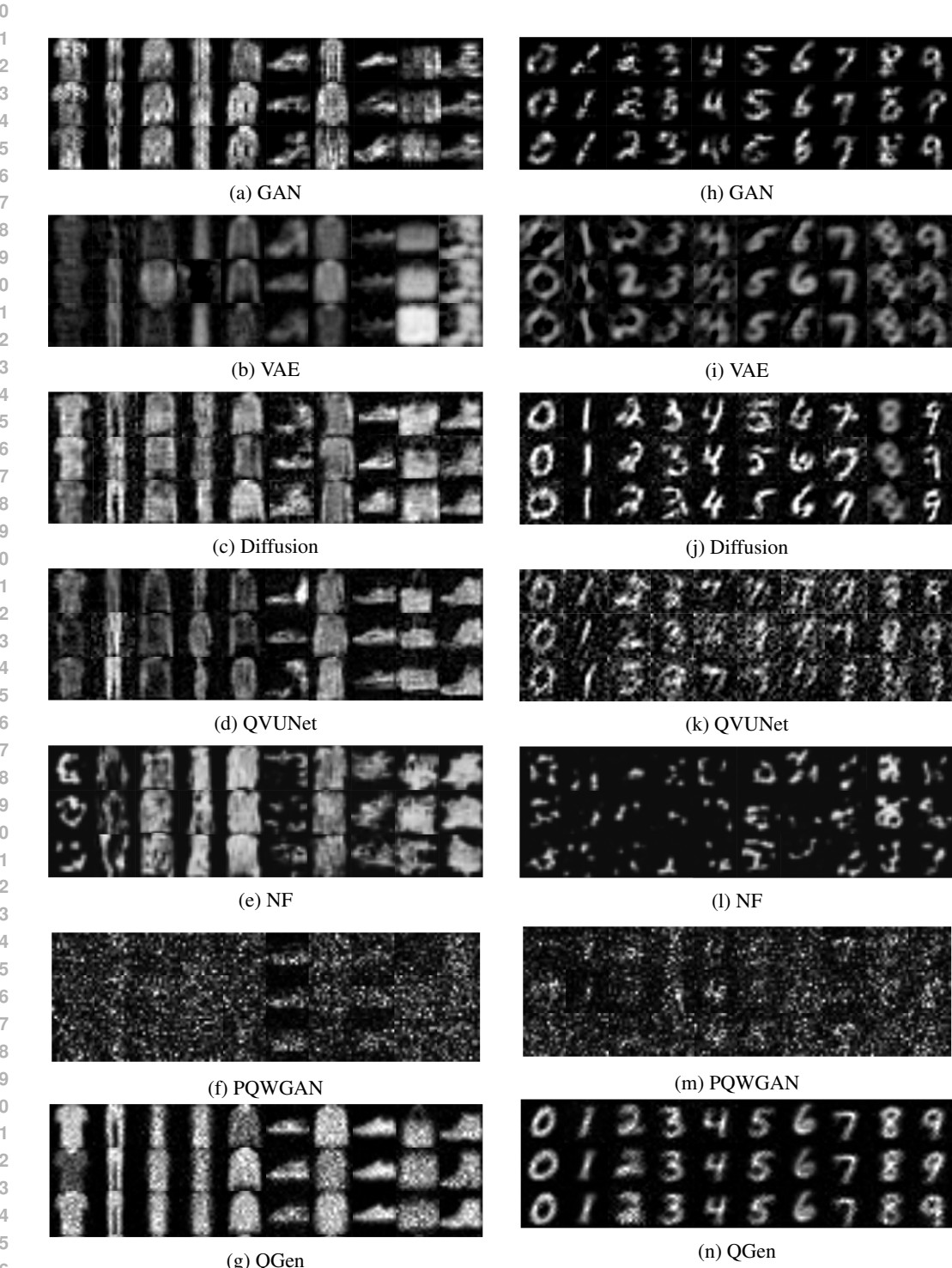

Figure 12: Comparison of generated samples across different models on FashionMNIST (left) and MNIST (right).

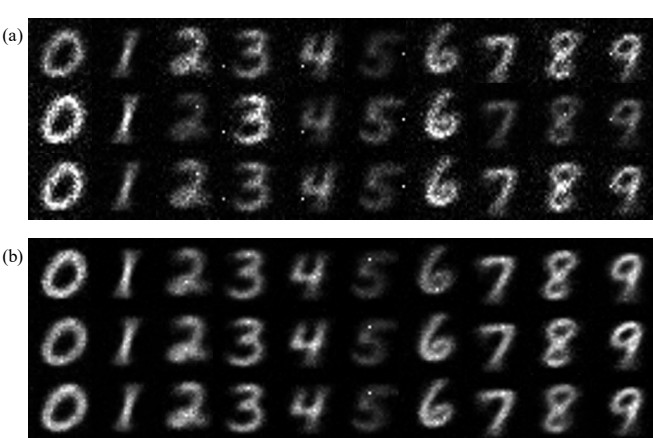

Figure 13: Generated MNIST samples ($28 \times 28$). (a) Circuit 1 ($N_\theta = 3780$); (b) Circuit 3 ($N_\theta = 18,540$). Circuit 3 produces clearer digits across all classes, showing improved generation with higher model capacity.

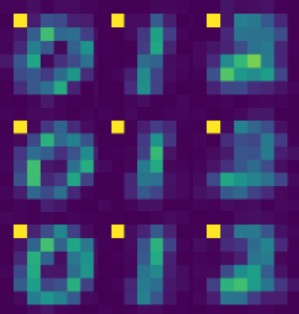

Figure 14: **Class-conditional generation on MNIST (8×8) for digits 0–2.** Circuit 1, $N_\theta = 2,472$, $l_0 = 12$.

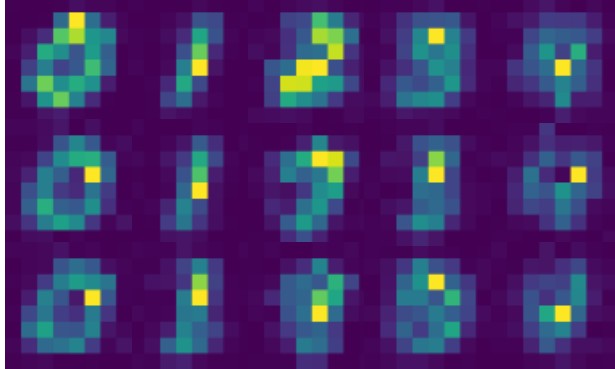

Figure 15: Class-conditional generation on MNIST (8×8) for digits 0-4. Circuit 1, $N_\theta = 2,472$, $l_0 = 12$.

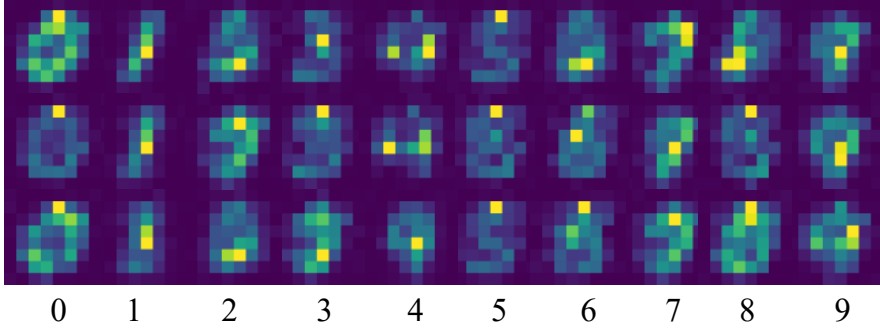

0    1    2    3    4    5    6    7    8    9

Figure 16: Class-conditional generation on MNIST (8×8) with Circuit 2. Trained on digits 0–9 using $N_\theta = 12{,}360$. Shows reasonable generation across all classes.

