# OpenReview forum: "A purely Quantum Generative Modeling through  Unitary Scrambling and Collapse"
_ICLR.cc/2026/Conference — Submitted to ICLR 2026_

### Official Review · Reviewer_CurE · 2025-10-21

**Soundness:** 2
**Presentation:** 3
**Contribution:** 2
**Rating:** 4
**Confidence:** 4

**Summary:**

This paper introduces QGen, a novel framework for generative modeling that is purely quantum-native as claimed, implemented with gate-based quantum circuits. The model is conceptually inspired by classical diffusion models but proposes a unique correspondence on quantum hardware. Its forward process involves repetitive cycles of: 1) amplitude encoding of input data perturbed by Gaussian noise, and 2) unitary delocalization to scramble the information and establish quantum correlations. The reverse, denoising process is performed by a series of pre-trained parameterized quantum circuits (PQCs), which are trained to refocus the scrambled state to reconstruct the original data distribution from a simple Gaussian prior.

**Strengths:**

*Novel Quantum-Native Scrambling Process*: The combination of Gaussian noise injection with unitary delocalization is a key innovation. It constructs a tractable Gaussian prior akin to classical diffusion models, while the unitary component ensures the path to this prior is highly expressive and leverages quantum entanglement, creating richer perturbation modes than classical noise alone.

*Comprehensive Empirical Analysis*: The paper provides a thorough ablation study comparing different noise injection strategies (purely unitary, Gaussian-only, combined) and rigorously analyzes performance under finite-shot sampling. This hardware-aware evaluation is crucial for establishing the model's practical feasibility in the NISQ era.

*Effective Training Strategy*: The layer-wise, measurement-based training of the collapse circuits effectively decomposes the challenging inverse problem into tractable sub-tasks. This approach is a pragmatic and well-motivated solution to mitigate the barren plateau problem that plagues the training of deep quantum circuits.

*Theoretical Analysis*: The paper provides theoretical justification for its design choices, notably arguing for the sufficiency of unitary reverse steps to invert the delocalization effect.

**Weaknesses:**

*Heavy Dependence on Amplitude Encoding*: The heavy reliance on amplitude encoding at every step, without a prior assumption of the distribution, constitutes a major bottleneck. Amplitude encoding is computationally expensive and notoriously difficult to implement on real gate-based hardware for arbitrary, non-simple distributions. This significantly undermines the claim of near-term feasibility and poses a fundamental scalability challenge.

*Not a Fully Coherent Quantum Evolution in the Forward Process*: The Gaussian diffusion channel relies on a destructive projective measurement and classical noise addition followed by later amplitude encoding, creating a "quantum-classical-quantum" reset at each step. This nuances the claim of a "purely quantum" process and imposes fundamental bottlenecks for scaling.

*Limited Demonstration of Scalability*: The experiments are confined to low-resolution grayscale images (e.g., 16×16). It remains an open and critical question whether the framework can scale to more complex, high-dimensional data due to the reasons mentioned above.  While the reviewers understands the difficulity on the software side, but I believe it can be more promising even if the resolution is doubled  (32 * 32) and more complex data is used.

**Questions:**

I do not have further questions. The method is clear to me.

**Details Of Ethics Concerns:**

No.

---

> ### Author Response · Authors · 2025-11-24
>
> We sincerely thank the reviewer for the precise and knowledgeable assessment. We appreciate that you recognized the "novel quantum-native scrambling" and the "effective training strategy" as key strengths. Below we address the concerns regarding amplitude encoding, "Quantum-Classical-Quantum" reset, and scalability.
>
> (w1-**Dependence on Amplitude Encoding**) We appreciate the reviewer’s concern regarding the reliance on amplitude encoding. You are correct that for arbitrary distributions, this constitutes a computational bottleneck with exponential scaling. However, we respectfully clarify that QGen does not operate in this worst-case regime. Real-world data distributions, such as images, occupy a low-dimensional manifold, allowing for approximation by polynomial-depth circuits rather than the exponential depth required for worst-case states.
> Crucially, the perceived dependence on amplitude encoding at every step is an artifact of the current simulation workflow, not an intrinsic requirement of the framework. On physical hardware, Gaussian perturbations can be natively implemented through well-established mechanisms such as Gaussian-randomized coherent rotations or stochastic unitary ensembles [1, 2]. These methods physically realize diffusive Gaussian dynamics directly on the Hilbert space without breaking quantum coherence [3]. This coherent implementation preserves the quantum state throughout the trajectory, thereby eliminating the need for intermediate measurements and the subsequent costly re-encoding steps. Consequently, amplitude encoding is required only for the initial state preparation rather than iteratively, significantly mitigating the cumulative scalability bottleneck.
>
> (w2-**"Quantum-Classical-Quantum" Reset**) We appreciate the reviewer’s keen physical observation regarding the apparent “quantum–classical–quantum” reset in the simulated forward process. However, we respectfully clarify that this measurement–re-preparation loop is a feature of the simulation workflow, not a fundamental physical requirement of the QGen algorithm.
> As detailed in our response to W1, Gaussian perturbations can be implemented natively as coherent quantum operations, theoretically supporting an end-to-end unitary trajectory that eliminates the need for intermediate projective measurements. Thus, the observed "reset" is a simulation artifact, not an intrinsic algorithmic constraint. From a mathematical perspective, while our simulation utilizes a measurement loop for implementation, the resulting transformation constitutes a valid CPTP map and is fully interpretable as a mixed-unitary channel within the framework of open quantum systems. Unlike hybrid models that collapse quantum states into classical vectors for neural processing, our evolution describes a stochastic quantum trajectory defined strictly within the Hilbert space density matrix formalism. Finally, the designation "purely quantum" serves to distinguish QGen from hybrid architectures; in our framework, the classical computer functions solely as a control plane for parameter updates, while the data flow itself resides entirely in the quantum domain.

---

> > ### Author Response · Authors · 2025-11-24
> >
> > (w3-**Scalability and Resolution**) We appreciate the reviewer’s constructive push for higher resolution demonstrations and the insightful observation regarding the difficulties on the software side. To directly address your suggestion to double the resolution toward the $32 \times 32$ regime, we extended our evaluation to full resolution $28 \times 28$ MNIST images as detailed in Appendix C. In these experiments, QGen maintained stable training dynamics and competitive generation quality, validating that the scrambling–collapse mechanism successfully scales to this complexity without requiring exponential resources. This result is particularly encouraging given that many hybrid models still restrict evaluations to heavily downsampled $8 \times 8$ grids. Furthermore, we fully align with your view that the framework holds even greater promise as the software stack matures. We agree that both quantum hardware and software infrastructure, particularly regarding PQC design and optimization strategies, possess immense potential for further evolution. Our framework is designed to be modular and will naturally benefit from these community wide advancements. As these underlying technologies improve, we anticipate that QGen will exhibit significantly enhanced capability and efficiency. Consequently, scaling to even more complex domains remains the central focus of our ongoing and future research, and we believe the current results provide the necessary empirical foundation for that trajectory.
> >
> > We sincerely appreciate your rigorous assessment. Your critical feedback, particularly regarding the physical coherence of the forward process and scalability, motivated us to provide deeper theoretical clarifications and explicitly highlight our empirical validation in the full resolution regime. We believe these efforts establish QGen as a robust contribution to the field, and we thank you for guiding us toward a significantly stronger and more rigorous manuscript.
> >
> > [1] Onorati, E., Buerschaper, O., Kliesch, M., Brown, W., Werner, A. H., & Eisert, J. (2017). Mixing properties of stochastic quantum Hamiltonians. Communications in Mathematical Physics, 355(3), 905-947.
> >
> > [2] Wallman, J. J., & Emerson, J. (2016). Noise tailoring for scalable quantum computation via randomized compiling. Physical Review A, 94(5), 052325.
> >
> > [3] Weedbrook, C., Pirandola, S., García-Patrón, R., Cerf, N. J., Ralph, T. C., Shapiro, J. H., & Lloyd, S. (2012). Gaussian quantum information. Reviews of Modern Physics, 84(2), 621-669.

---

> ### Comment · Reviewer_CurE · 2025-11-26
>
> I thank the reviewer for providing more detailed replies. I still have concerns from the following perspectives: 1) heavy use of amplitude encoding: even though real-world data distributions such as images may occupy a low-dimensional manifild, it does not automatically mean that the repeated amplitude encoding process is a trivial process. This argument, from my perspective, is weak without support of empirical evidence; 2)  the quality of the generated images is still relatively poor. Therefore, I would maintain my original score.

---

> > ### Author Response · Authors · 2025-11-27
> >
> > We thank the reviewer for the additional comments and fully respect the concerns raised. We would like to clarify two points to avoid confusion about the scope and intent of our contribution.
> > 1. Amplitude Encoding: the "repeated encoding" bottleneck is a simulation artifact, not a physical requirement.
> >
> >      – Frequency. On hardware using coherent noise injection, amplitude encoding occurs only once at $t=0$. Intermediate steps are fully unitary with no re-loading.
> >
> >      – Modularity. Efficient data encoding remains a universal challenge across QML. QGen does not rely on amplitude encoding as a core mechanism and is compatible with alternative, more scalable encodings as they emerge.
> >
> > 2.Image Quality Context
> >
> >    – Resolution context. Most quantum generative models restrict experiments to $8 \times 8$  or similarly downsampled grids. Our $28 \times 28$ results therefore demonstrate feasibility at a substantially higher resolution.
> >
> >    – Image quality. Empirically, QGen achieves noticeably higher performance than both hybrid and classical baselines evaluated under the same parameter and resolution constraints.
> >
> > We thank the reviewer again for the constructive feedback and will incorporate these clarifications into the revised manuscript.

---

### Official Review · Reviewer_aEzc · 2025-10-26

**Soundness:** 3
**Presentation:** 4
**Contribution:** 3
**Rating:** 8
**Confidence:** 4

**Summary:**

This manuscript proposes the Quantum Scrambling and Collapse Generative Model (QGen) for generative machine learning of classical data with a pure quantum paradigm. Similar to the classical diffusion model, it involves a forward diffusion process and a backward denoising process. In the forward process, a Gaussian diffusion channel and a scrambling unitary is applied to the input data, and in the backward side, a parameterized unitary is applied and optimized through loss function minimization. In numerical experiments, they show the effectiveness of the model in generating MNIST data. They further show its advantage over classical diffusion model and a hybrid quantum model with comparable parameter budgets, and the necessity of both Gaussian channel and scrambling unitary in the forward process.

**Strengths:**

Overall, the manuscript is well-written and could become an interesting contribution to the development of quantum diffusion model for generative learning, with the caveat for some clarifications. The manuscript also provides comprehensive and detailed numerical study to provide theoretical insights and demonstrate the effectiveness and advantage of this model in the generative task. Given these clarifications in an author response, I would be willing to increase the score.

**Weaknesses:**

The major concern is the unitary scrambling in the forward process. The Gaussian diffusion channel simply mimics the process of injection of Gaussian white noise to data in classical diffusion process. The unitary instead scrambles the input state from prior channel. The intuition for this unitary is not clear to us. Through the comparison in Fig. 3, both U-sched, G-only and QGen seems to converge to a stable measurement statistics.

**Questions:**

In Fig. 3, is there any theoretical or numerical justifications on whether the converged measurement statistics of QGen is a Gaussian-like distribution.

In Table 2, it seems that T=8 leads to the best performance with different setup, is there any specific reason for it?

In this manuscript, the encoding of classical data to quantum states is implemented via amplitude encoding, which is in principle challenge. For example, in Eq. (4), the amplitude embedding map of the diffusion channel could lead to a highly entangled state which is hard to prepare. We hope the authors provide comments and statements for this limitation.

In the statement below Eq. (5), the authors states that the unitary evolution enhances expressivity through the introduction of non-classical randomness through coherent evolution. However, there is indeed no non-classical randomness injected into the dynamics as once the unitary is chosen, the quantum state evolves in a deterministic way. The authors should clarify it.

---

> ### Author Response · Authors · 2025-11-24
>
> We thank the reviewer for the thoughtful, constructive, and encouraging review. We appreciate the positive assessment regarding the clarity of our presentation, the soundness of the proposed framework, and the comprehensiveness of the numerical study. We are particularly grateful for the reviewer’s recognition of QGen as an interesting contribution to the development of generative learning. The precise comments provided have helped us significantly strengthen the conceptual intuition and terminological rigor of the manuscript.
>
> (W-**Intuition of Unitary Scrambling**) We appreciate the opportunity to clarify the physical intuition behind the unitary scrambling mechanism. The reviewer is correct that the Gaussian channel $\mathbf{N_{\beta_t}}$ alone drives the system toward a Gaussian prior. However, the unitary delocalization operator $\mathbf{U_{\theta_t}}$ is not to ensure convergence, but to shape the dynamical trajectory, which is crucial for enabling the reversibility learned by our backward collapse process.
>
> To intuit this, consider the analogy of a drop of ink in water. Under simple diffusion, the ink spreads smoothly via countless uncorrelated Brownian fluctuations. While the final distribution appears simple, the underlying path is highly irregular and essentially impossible to retrace (the "G-only" baseline). In contrast, vigorously stirring the water after injection introduces large-scale mixing patterns (folding and stretching) that organize the diffusion into a structured and globally correlated process. This mirrors the quantum setting, where the unitary operator $\mathbf{U_{\theta_t}}$ acts as a coherent mixer. It transforms local Gaussian perturbations into global, entangled correlations. This coherent structure allows the parameterized reverse circuit to learn a unified dynamical inversion rather than attempting to recover innumerable independent noise events. This intuition is quantitatively supported by our ablation results (Table 2). We will revise Section 3.1 to explicitly summarize this physical interpretation.

---

> > ### Comment · Reviewer_aEzc · 2025-11-27
> > **Reply to the Comment for weakness**
> >
> > We thank authors' comment on the weakness for clarifying the physical intuition of the unitary scrambling.
> > We still have two concerns as follows.
> > 1. What is the reason for the preference of the converged Gaussian-like distribution over others?
> > 2. For the Unitary delocalization operator, which is a unitary circuit consisting of local rotations, does it indicate that this step can also be implemented on the classical computer efficiently instead of a quantum circuit? If so, then it seems that the whole forward process can be implemented fully classically, and we hope that the authors should clarify this point.

---

> > > ### Author Response · Authors · 2025-11-27
> > >
> > > We thank the reviewer for the thoughtful follow-up questions. These points help
> > > clarify both the design choices in the forward process and the quantum nature of the delocalization operator. We address each question below.
> > >
> > > (Q1) Regarding the rationale for using Gaussian perturbations, we clarify the following. We acknowledge that our initial choice of Gaussian perturbations follows the standard practice in classical generative modeling, where Gaussian noise serves as a stable and well-behaved diffusive prior. While the choice is empirical in origin, it aligns closely with the theoretical structure developed in our work.
> > >
> > > As outlined in our responses to other reviewers, we formalize QGen as a thermodynamic recovery problem governed by two constraints: (i) preservation of coherent information for recoverability, and (ii) maintenance of a non-degenerate Fisher Information Matrix for learnability. These conditions define a critical chaotic window in which the forward trajectory must remain. Gaussian-like distribution interacts naturally with these requirements.
> > >
> > > From the perspective of recoverability, Gaussian perturbations provide a controlled and smooth increase of entropy, avoiding the abrupt thermalization that can occur with more irregular or discontinuous noise processes. Moreover, Gaussian distributions maximize entropy subject to a fixed variance, which yields a high-diversity yet energy-constrained prior. In practice, this helps keep the forward trajectory within a regime where the coherent information does not collapse too rapidly, supporting the recoverability.
> > >
> > > From the geometric perspective, Gaussian noise generates a smooth, unimodal, and
> > > well-conditioned evolution of the Born distribution, which leads to a more stable
> > > distributional loss landscape. This ensures that the phase-invariant solution manifold remains geometrically accessible, preventing the formation of singular Jacobians or optimization barriers often associated with discrete or multimodal priors[1]. Unlike featureless white noise, the Gaussian distribution retains a specific covariance structure that acts as a "breadcrumb trail" for the Fisher Information, ensuring that gradients remain informative [2]. As a result, the optimization landscape remains accessible within the learnable regime required.
> > >
> > > Thus, Gaussian noise provides a natural and compatible choice for QGen’s forward process, though not the only possible one. Exploring non-Gaussian alternatives that satisfy the QGen conditions remains an interesting direction for future work. We will clarify this in the revised manuscript.
> > >
> > > (Q2) We appreciate the reviewer’s careful question. Although each gate in the delocalization operator is a single-qubit rotation and therefore trivial to simulate in isolation, its action in QGen is applied to the full $2^n$-dimensiona amplitude vector of an $n$-qubit quantum state. Even when composed solely of local rotations, the resulting unitary acts globally on the Hilbert space and induces coherent interference among the $2^N$ amplitudes, effectively  functioning as a high-dimensional basis rotation.
> > >
> > > Classically reproducing the same transformation would require explicitly storing and updating all $2^n$ amplitudes (or equivalently the full Born probability table), which incurs exponential overhead. In contrast, the quantum circuit performs this evolution natively, without materializing the full state vector. Moreover, the coherent nature of the quantum evolution is essential for QGen: it preserves the reversible structure needed to maintain coherent information during the forward process, which is required for the recovery condition in our theoretical framework. A fully classical forward process would lack this coherence and would reduce to a Markovian diffusion, whose long-time limit rapidly erases the correlations needed for inversion.
> > >
> > > Thus, while the gate primitives themselves are local, their composition acts on an exponentially large quantum state and cannot be simulated efficiently by a classical algorithm without enumerating the entire state vector. We will clarify this distinction in the revised manuscript.
> > >
> > > We appreciate the reviewer’s continued engagement. We will incorporate these clarifications into the revised manuscript to improve the presentation and clarify the scope of our claims.
> > >
> > > [1] Dauphin, Y. N., Pascanu, R., Gulcehre, C., Cho, K., Ganguli, S., & Bengio, Y. (2014). Identifying and attacking the saddle point problem in high-dimensional non-convex optimization. *Advances in neural information processing systems*, *27*.
> > >
> > > [2] Martens, J. (2020). New insights and perspectives on the natural gradient method. *Journal of Machine Learning Research*, *21*(146), 1-76.

---

> > > > ### Comment · Reviewer_aEzc · 2025-11-28
> > > > **Reply to Official Comment by Authors for Weakness**
> > > >
> > > > We appreciate the authors' detailed reply, and we think that the weakness is well addressed now. We would like to raise the score of soundness from 3 to 4.

---

> > > > > ### Author Response · Authors · 2025-11-28
> > > > >
> > > > > We sincerely thank the reviewer for the positive feedback and for raising the soundness score. We are glad that our clarifications regarding the theoretical framework and physical intuitions have addressed the concerns. We will incorporate the discussed theoretical explanations and experimental details into the revised manuscript.

---

> ### Author Response · Authors · 2025-11-24
>
> Regarding your questions:
>
> (Q1-**Gaussian-like Statistics**) We thank the reviewer for this insightful question regarding the measurement statistics. There are both theoretical and numerical justifications supporting our statement that the converged measurement statistics exhibit Gaussian-like behavior.
>
> Theoretically, the stochastic part of each forward step applies an explicit Gaussian perturbation to the classical amplitude vector(Eq. (3)), while the subsequent unitary delocalization preserves the norm and coherently redistributes this noise across the state space. Repeating this process drives the system toward a stationary distribution whose first and second moments follow Gaussian scaling, even though the measurement distribution is discrete.
>
> Numerically, to validate this beyond visual inspection, we conducted a rigorous statistical evaluation. Across 5920 generated histogram instances, we applied both Shapiro–Wilk and D’Agostino–Pearson normality tests. The results show an 88.70 pass rate for joint normality, with an average Shapiro–Wilk p-value of 0.6323  and an average D’Agostino–Pearson p-value of 0.2413. These results provide strong empirical support that the converged measurement statistics are well-approximated by a Gaussian profile in practice. We have added a summary of this statistical analysis to Appendix to quantitatively justify the Gaussian assumption.
>
> (Q2-**Optimality of T=8 (Table 2)**) We appreciate the observation regarding the performance peak at $T=8$. This optimum reflects a trade-off between reversibility and optimization difficulty.
>
> Theoretically, when T is too small, each scrambling step becomes overly strong, which quickly destroys structural information and makes the reverse process harder to learn. Increasing T smooths the perturbations through the variance schedule, allowing information to be preserved in a more controlled way. However, when T becomes large, the cumulative quantum perturbations increasingly mix the state and push it toward a maximally mixed or highly entangled state, leaving less recoverable structure for the reverse circuit to reconstruct. This explains the improvement from small to intermediate T and the degradation at very large T in the scheduled variants (U-sched and QGen).
>
> For the U-fixed variant, each step applies an independent and uncontrolled perturbation, so increasing T simply drives the system toward a more mixed state, resulting in monotonic degradation.
>
> In practice, the optimality of T=8 also reflects mild error accumulation during backward inversion and the random sampling of the delocalization parameters $\theta_t$. We expect the best value of T to vary with architecture and hardware constraints, but in our setting T=8 provides the most effective trade off.
>
> (Q3-**Complexity of Amplitude Encoding**)The reviewer raises a valid point regarding the challenge of amplitude encoding for highly entangled states. We acknowledge that exact amplitude encoding for arbitrary distributions scales exponentially ($O(2^n)$). In our experiments, the data dimension and qubit count remain small enough that state preparation is tractable in practice. We will also clarify that amplitude encoding serves only as the data-loading mechanism, and QGen is fully compatible with more hardware-efficient encoding or state-preparation techniques as they become available. We will explicitly discuess this limitation in the revision.
>
> (Q4-**Clarifying "non-classical randomness"**) We thank the reviewer for highlighting this phrasing. We agree that the unitary evolution is technically deterministic once the parameters are fixed. Our intent was to emphasize that a deterministically applied but stochastically parameterized unitary encourages the system to explore states characterized by genuinely non-classical correlations (e.g., multipartite entanglement). Unlike classical stochastic noise, which lacks coherent interference and cannot generate entanglement, this process leverages coherent evolution and quantum interference to generate complex state patterns and global dependencies. The benefit is therefore not "randomness" per se, but the enhanced expressivity and scrambling capability arising from the induced non-classical structure. In the revision, we will replace "non-classical randomness" with "randomly parameterized unitary dynamics" to accurately reflect this mechanism.
>
> We sincerely appreciate your endorsement and the high quality feedback provided. Your insights, particularly regarding the physical intuition of unitary scrambling and the terminological precision of quantum randomness, have driven us to significantly refine the manuscript's clarity and rigor. We are fully committed to integrating these revisions to ensure the final version serves as a robust contribution to the field of quantum generative learning.

---

> > ### Comment · Reviewer_aEzc · 2025-11-27
> > **Reply to Comments for questions**
> >
> > The authors' comments carefully handled and addressed the questions.

---

### Official Review · Reviewer_AViQ · 2025-10-26

**Soundness:** 3
**Presentation:** 2
**Contribution:** 2
**Rating:** 4
**Confidence:** 3

**Summary:**

The paper introduces the VAE-like quantum generative model with quantum scrambling and collapse where scrambling process utilizes Gaussian diffusion channels and unitary delocalization and collapse process employs parameterized quantum circuits to reform the scrambled distributions into structured outputs.

**Strengths:**

The topic is interesting. It explore the quantum operations to generalize the  Besides, the paper presents extensive numerical experiments that provide empirical support for the proposed method.

**Weaknesses:**

1. The scalability of the proposed method is not explicitly discussed, leaving uncertainty about its feasibility for larger systems.
2. While the proposed method demonstrates promising empirical performance, the paper lacks a concrete rigorous theoretical analysis elucidating its advantages over prior classical or quantum approaches.

**Questions:**

1. The caption of Fig.2 is a little bit confused. For instance, there is no $\mathcal{C}_{\phi}$ in figure and what's the concrete presentation of such parameterized collapse operator?
2. Could the authors clarify what type of projective measurement is performed during the scrambling process described in line 160? Does each element of the probability vector $p_{t-1}$ correspond to the probability of a specific projective measurement outcome? Moreover, is the process of adding Gaussian noise purely classical? what is the rationale for selecting Gaussian noise rather than quantum noise?
3. As the proposed scheme involves performing projective measurements at every step $t$, how does this affect the overall scalability and feasibility of the approach for larger systems?
4. Is the amplitude embedding process of the noise distribution efficiently?
5. In line 276, since each step $t$ requires a measurement in the computational basis, does this not lead to an exponential measurement cost?
6. Since Equation 7 uses the KL-divergence to quantify the distance between distributions, could the authors clarify whether the distributions ($\tilde{p}_{\rho_{t-1}}\|\tilde{p}_{\tilde{\rho}_{t-1}}$) at steps have a non-negligible overlap? If the overlap is negligible, how might this affect the reliability of using KL-divergence as the distance metric?
7. How is the number of steps $T$ determined, and how does this choice influence the model’s performance?
8. What is the concrete quantum-circuit architecture used to implement $U_{\theta_t}$ (the unitary delocalization operator) in line 170?

---

> ### Author Response · Authors · 2025-11-24
>
> We appreciate the reviewer’s thoughtful evaluation and the recognition of the strengths of our submission, particularly the novelty of the quantum scrambling–collapse architecture and the depth of the empirical study. The detailed feedback is extremely helpful, and we address each point below.
>
> (W1-**Scalability of the Method**) We appreciate the reviewer’s constructive comment and agree that a dedicated discussion on scalability was necessary to fully contextualize the method's potential.To address this uncertainty, we have revised the manuscript to explicitly evaluate the feasibility of QGen for larger systems across both algorithmic and hardware dimensions.
>
> From an algorithmic perspective, QGen is designed so that training remains tractable within the practical limits of current NISQ devices. By formulating the entire generative trajectory directly in the quantum domain, QGen avoids the communication and synchronization bottlenecks of hybrid pipelines. The stepwise objective decomposes the global generative task into locally conditioned subproblems, which reduces the circuit depth requirement from the prohibitive $\Omega(n)$ layers to only $O(\log n)$ layers per timestep, thereby mitigating the barren plateau problem inherent to deep quantum circuits. We further validated scalability on full-resolution 28x28 images, as detailed in Appendix C.
>
> Regarding hardware feasibility, we conducted dedicated experiments to evaluate the interaction between quantum noise and finite-shot sampling. Across a range of depolarizing and amplitude-damping noise strengths (Table 2.1), we observe a distinct stability plateau. Furthermore, scaling analysis (Table 2.2) demonstrating that QGen operates without exponential resource explosion. While NISQ hardware unavoidably limits absolute scale, our empirical findings consistently show that QGen maintains stable behavior and competitive generative quality within these constraints.
>
> (W2-**Theoretical Analysis**) We appreciate the reviewer’s emphasis on greater theoretical rigor. We agree that elucidating the formal advantages of our approach over prior methods is essential. We model QGen as a thermodynamic recovery problem operating within a critical window: scrambling must be sufficient to avoid a trivial ansatz, but limited to prevent the vanishing of Fisher Information.
>
> The framework is built upon three foundational concepts. The Forward Scrambling Operator ($D$) describes unitary evolution under a chaotic Hamiltonian $H(\tau)$ that delocalizes information into global entanglement and effectively thermalizes local subsystems. Coherent Information ($I_c$) establishes the fundamental boundary for information preservation, where $I_c(R \rangle D') \equiv S(\rho_{D'}) - S(\rho_{RD'})$ quantifies the source's retrievability. The Distributional Loss Function ($\mathcal{L}(\boldsymbol{\theta})$) targets only the Born probability distribution $P_{\boldsymbol{\rho}}(x)$, introducing a strategic relaxation in the solution space.
>
> Building upon this, we adopt two foundational theorems. The Conservation of Generative Potential establishes the thermodynamic advantage, demonstrating that unlike irreversible thermalization, unitary scrambling preserves the required coherent information ($I_c \ge S(P_{\text{data}})$), providing a rigorous condition for the existence of a successful inverse map. The Geometry of Learnability provides the computational advantage: by targeting probabilities rather than pure states, the method exploits a phase invariant solution manifold that dramatically reduces optimization complexity, provided the system operates where the gradient variance remains non vanishing.
>
> These advantages are unified by the Fisher Coherent Conjecture, which defines the optimal operational point $t^*$ by maximizing the Fisher Coherent Product $\mathcal{J}(t)$:$\mathcal{J}(t) = I_c(t) \cdot \left[ \det(\mathcal{F}(t)) \right]^{1/k}$.  Experimental results demonstrate that our model successfully operates within critical zone, where global quantum correlations are thermodynamically recoverable (based on $I_c$) and the solution manifold is geometrically accessible (based on $\det(\mathcal{F})$) to efficient gradient optimization.

---

> ### Author Response · Authors · 2025-11-24
>
> Regarding your questions:
>
> (Q1-**Fig.2 Caption & Collapse Operator**) We appreciate your careful observation regarding the ambiguity in Figure 2.We clarify that the parameterized collapse operator $C_\phi$ corresponds to the variational quantum circuit used to invert the diffusion. In the figure, the purple blocks labeled $C_t$ represent the specific Parameterized Quantum Circuit (PQC) applied at each timestep. We have updated the caption and the method section in the revision to explicitly label these components and link them to the detailed architecture provided in Appendix A.
>
> (Q2-**Projective Measurement & Noise Rationale**) We appreciate the reviewer’s insightful questions. Regarding the projective measurement and the probability vector: your understanding is correct. In every scrambling step $D_t$, we perform a projective measurement in the computational basis. Consequently, each element $p_{t-1}(i)$ of the probability vector explicitly corresponds to the probability of observing the specific basis state $|i\rangle$, calculated via the Born rule: $p_{t-1}(i) = \langle i|\rho_{t-1}|i\rangle$.
>
> Regarding the noise rationale: We acknowledge that in our current experimental simulations, the addition of Gaussian noise is implemented as a classical injection strategy. However, this distribution is fundamentally compatible with a native quantum framework. On physical hardware, Gaussian perturbations can be natively implemented through well-established mechanisms such as Gaussian-randomized coherent rotations or stochastic unitary ensembles[1,2]. These methods physically realize diffusive Gaussian dynamics directly on the Hilbert space without breaking quantum coherence, consistent with the formalism of Gaussian quantum information [3].
>
> We selected Gaussian perturbations because they provide a tunable, analytically tractable entropy schedule and combine cleanly with unitary scrambling. In contrast, native quantum noise channels (e.g., depolarizing or amplitude damping) converge to fixed points that rapidly destroy recoverable structure, making them unsuitable as a generative prior. The empirical results in Fig.3, Fig.5, and Table.2 further support this design choice by demonstrating that controlled Gaussian perturbations lead to significantly improved performance. The revised manuscript now clarifies both the practical and conceptual motivations for this design choice.
>
> (Q3,Q5-**Measurement Cost and Scalability**) We appreciate the reviewer’s concern about the impact of stepwise projective measurements on scalability and measurement cost. QGen can requires only the statistics that appear in the loss, which can be directly approximated from a finite number of measurement samples. In this sense, QGen follows the standard generative-modeling paradigm in which empirical samples provide unbiased estimators of the required expectations, allowing us to bypass the prohibitive cost of full state tomography. Empirically, Table 2.2 demonstrates that the required number of measurements does not scale exponentially with the Hilbert-space dimension in the regime we study. Unlike classical diffusion models requiring $T \approx 1000$ steps, QGen achieves high fidelity with a short trajectory of $T \approx 8$. Since training decomposes into local subproblems, the total measurement overhead scales linearly with $T$, ensuring the aggregate cost remains computationally tractable without exponential explosion
>
> (Q4-**Amplitude Encoding Efficiency**)We acknowledge that exact amplitude encoding for arbitrary distributions is exponentially hard. However, QGen does not operate on worst-case random distributions. The target and prior states in our setting are structured and sparse, occupying a low-dimensional manifold, which allows them to be approximated by polynomial-depth circuits [4, 5]. Furthermore, Amplitude encoding is used solely as an initial data-loading mechanism and is not fundamental to the theoretical construction of QGen. The framework is compatible with any improved state-preparation technique that may become available in the fault-tolerant era, which would directly strengthen its practicality.

---

> ### Author Response · Authors · 2025-11-24
>
> (Q6-**KL Divergence & Overlap**) This is an excellent technical point. You are correct that negligible overlap leads to KL instability. We ensure stability and reliable gradient signals through three complementary mechanisms. First, the forward process acts as a guide; since the backward process is trained to reverse it step by step, the generated distribution ($p_{\rho_{t-1}}$) naturally maintains support close to the target ($p_{\tilde{\rho}_{t-1}}$), preventing complete divergence. Second, we employ a hybrid loss function (Equation (7)) that includes an $L_1$ distance term alongside the KL divergence. This $L_1$ component is significantly more robust to near-disjoint supports and provides essential local gradient signals, guiding the model toward convergence even when the KL term is minimal. Furthermore, for numerical conditioning, we employ a renormalization strategy as defined in Equation (8) and Figure 4a. This divides the measurement probabilities by their maximum value, effectively rescaling the probability landscape to compensate for exponential sparsity and providing a better-conditioned basis for divergence calculations.
>
> (Q7-**Determination of Timesteps T**) We appreciate the reviewer's specific question regarding the determination of the timestep hyperparameter $T$. The choice of $T$ balances forward scrambling granularity and reverse reconstruction difficulty. If $T$ is too small, aggressive perturbations destroy structural information. If $T$ is too large, the cumulative unitary delocalization drives the state toward a maximally mixed state, resulting in irreversible information loss. Furthermore, a longer trajectory exacerbates error accumulation during the iterative reverse process. As shown in Table 2, we determined empirically that $T=8$ provides the optimal trade-off between stability and accuracy for our architecture.
>
> (Q8-**Unitary Delocalization Architecture**) We appreciate the reviewer's specific inquiry regarding the architectural realization of the scrambling unitary. As outlined in the methodology, the unitary delocalization operator $U_{\theta_t}$ is constructed from local rotations applied to qubits. To be concrete, in all our reported experiments, this is implemented as a sequence of three consecutive rotations ($R_z, R_x, R_z$) applied to each qubit. The rotation angles for these gates are independently sampled from a normal distribution $N(0, \sigma_t^2 I)$, where the variance is dynamically modulated by the noise schedule. While this specific $R_z-R_x-R_z$ decomposition is not the only possible choice, it was selected to optimize the balance between scrambling expressivity and hardware efficiency. This is particularly relevant for superconducting qubit systems, where $R_z$ gates are often implemented as virtual gates with near-zero duration and negligible error, rendering this sequence exceptionally friendly for NISQ-era deployment while maintaining the necessary capability to delocalize information.
>
> [1] Weedbrook, C., Pirandola, S., García-Patrón, R., Cerf, N. J., Ralph, T. C., Shapiro, J. H., & Lloyd, S. (2012). Gaussian quantum information. Reviews of Modern Physics, 84(2), 621-669.
>
> [2] García-Pérez, G., Rossi, M. A., & Maniscalco, S. (2020). IBM Q Experience as a versatile experimental testbed for simulating open quantum systems. npj Quantum Information, 6(1), 1.
>
> [3] Weedbrook, C., Pirandola, S., García-Patrón, R., Cerf, N. J., Ralph, T. C., Shapiro, J. H., & Lloyd, S. (2012). Gaussian quantum information. Reviews of Modern Physics, 84(2), 621-669.
>
> [4] Huang, H. Y., Broughton, M., Chen, M. C., Li, S., Mohseni, M., Babbush, R., ... & Neven, H. (2020). Power of data in quantum machine learning.*Nature Communications, 11*(1), 1-10.
>
> [5] Grover, L., & Rudolph, T. (2002). Creating superpositions that correspond to efficiently integrable probability distributions. *arXiv preprint quant-ph/0208112*.

---

> > ### Comment · Reviewer_AViQ · 2025-11-27
> >
> > Thank authors detailed answers. I still have the following questions:
> > 1. for Q3 and Q5, as you explained in answer of Q2, it requires perform the projective measurements and calculated via the Born rule via $p_{t-1}(i) = \langle i|\rho_{t-1}|i\rangle$. Thus, it requires get $2^n$ terms for all the computational basis. It seems have scalability issue.
> > 2. for Q7, is there any theoretical analysis about how to guide to determine such $T$ not only rely on empirical experiments?
> > 3. for W2, is there any specific theory analyze why the gradient variance remains non vanishing?

---

> ### Author Response · Authors · 2025-11-27
>
> We sincerely thank the reviewer for the rapid and deep engagement with our rebuttal. These follow-up questions hit precisely on the subtleties of quantum execution versus classical simulation. We provide the specific clarifications below.
>
> (Q1) We appreciate the reviewer's follow up regarding this important distinction. We clarify that the description provided in our previous response to Q2, where $p_{t-1}(i) = \langle i|\rho_{t-1}|i\rangle$, represents the theoretical probability distribution that physically governs the quantum system.  This expression specifies the probability model used in the loss formulation, but it does not imply that QGen evaluates all $2^n$ probabilities in practice.
>
> In a hardware realization, QGen requires only sampling access to this distribution. Instead of computing the full vector$\mathbf{p}_ {t-1}= ( p_{t-1}(i) )_ {i\in\{0,1\}^n}$, the algorithm uses a finite set of measurement outcomes drawn according to $p_{t-1}(i)$. The corresponding empirical frequency vector $\widehat{p}_ {t-1}(i) = \frac{N_i}{N_{\mathrm{shots}}}$, where $N_i$ is the number of times outcome $i$ is observed, serves as a Monte--Carlo approximation to the full distribution and is directly inserted into the loss. Moreover, Section~4.3 of the manuscript includes an empirical study of QGen under different shot budgets. Within the scales studied, the model exhibits stable generative behavior across a range of finite-shot settings.
>
> (Q2)We thank the reviewer for raising this important point. At present, we do not have a closed-form theoretical rule for determining the optimal number of timesteps $T$. This is consistent with the situation in classical diffusion models, where widely used choices such as $T=1000$ are also not derived from a sharp theorem but rather from a balance between forward information decay, reverse reconstruction difficulty, and computational cost.
>
> Nevertheless, our framework does suggest several principled theoretical directions for guiding the choice of $T$. On the forward side, one can analyze how scrambling and diffusion jointly affect information preservation, e.g., via the decay of coherent information $I_c(t)$ or the growth of entropy and Fisher-information–based quantities along the trajectory. This naturally suggests a “learnable window’’ in $T$ before the state becomes too close to a high-entropy, nearly featureless prior.  On the reverse side, larger $T$ increases the cumulative approximation and sampling error across steps, while too small a $T$ produces large reconstruction jumps. Thus $T$ is naturally governed by a trade-off between forward perturbation granularity and backward stability.
>
> Developing a more explicit theoretical characterization of this trade-off is an interesting open problem and a valuable direction for future work. In the current paper, we follow this theoretical picture as guidance to guide the search and then validate the choice empirically, with $T\in[8,10]$ consistently lying in the favorable regime for the present architecture and data resolution. We will clarify this discussion in the revised manuscript.
>
> (Q3)We thank the reviewer for requesting a more precise theoretical justification. The theoretical support for non-vanishing gradients in QGen arises from the interaction between structural mitigation through circuit decomposition and geometric mitigation resulting from phase invariance.
>
> Structurally, the primary constraint on trainability is the Barren Plateau phenomenon, where deep global circuits ($L \in \Omega(n)$) approximate unitary 2-designs, causing gradient variance to decay exponentially as $\text{Var}(\partial \mathcal{L}) \in \mathcal{O}(e^{-\alpha n})$[1].
>
> QGen avoids entering this regime by adopting a time-sliced decomposition of the generative trajectory. Rather than learning a single global unitary, each step optimizes a shallow parametrized circuit operating in a regime consistent with $L=\mathcal{O}(\log n)$. In such shallow-depth settings, cost-function–dependent barren-plateau theory predicts at most polynomial decay of gradient variance for local or structured objectives, $\mathrm{Var}(\partial_{\theta_k}\mathcal{L}) \in \Omega(\mathrm{poly}(n^{-1}))$, which supports practical trainability[2].
>
> Beyond circuit depth, the geometry of the loss landscape plays a critical role. Unlike state reconstruction which targets a specific pure state and requires precise phase alignment, our distributional loss targets the Born probabilities. This objective defines a phase-invariant solution manifold where directions corresponding to pure phase rotations possess zero curvature. By rendering the loss insensitive to these flat dimensions, the condition number of the effective Fisher Information Matrix is significantly improved compared to state-vector objectives[2], ensuring that the gradients remain not only non-vanishing but also effective for convergence. We will clarify this theoretical analysis in the revised manuscript.

---

> > ### Author Response · Authors · 2025-11-27
> >
> > We appreciate the reviewer’s continued engagement. We will incorporate these clarifications into the revised manuscript to improve the presentation and clarify the scope of our claims.
> >
> > [1] McClean, J. R., Boixo, S., Smelyanskiy, V. N., Babbush, R., & Neven, H. (2018). Barren plateaus in quantum neural network training landscapes. *Nature communications*, *9*(1), 4812.
> >
> > [2] Cerezo, M., Sone, A., Volkoff, T., Cincio, L., & Coles, P. J. (2021). Cost function dependent barren plateaus in shallow parametrized quantum circuits. *Nature communications*, *12*(1), 1791.

---

> ### Comment · Reviewer_AViQ · 2025-11-28
>
> Thanks for further clarification. Based on the detailed reply, most of my concerns are addressed and I would like to raise my score to 6.

---

> > ### Author Response · Authors · 2025-11-28
> >
> > We appreciate the reviewer’s time and careful reconsideration, and we thank them for the updated score and constructive feedback. Their comments have been genuinely helpful in improving the clarity of the work.

---

### Official Review · Reviewer_e3x3 · 2025-10-28

**Soundness:** 2
**Presentation:** 2
**Contribution:** 1
**Rating:** 2
**Confidence:** 5

**Summary:**

The authors propose a "pure" quantum framework for image synthesis, attempting to decompose the image into Gaussian-like noise and reconstruct the image by an inverse process represented by parameterized quantum circuits.

**Strengths:**

1. The use of the quantum scrambling process is interesting.
2. There are some ablation experiments investigated.

**Weaknesses:**

1. Typo in the citation (?Liu&…) in line 95 and line 107 → at least should check the basic format before submission. Same for "Appendix ??" in line 751.
2. Amplitude encoding circuit can be very deep, and extracting such information from n qubits means the amplitude/probability of each basis from 2^n space is required. Thus the required measurement shots is likely exponential with qubit count.
3. In abstract "a purely quantum paradigm that eliminates classical dependencies" → This description is written in the sense as if it is optimized in the quantum circuit like Grover-based method. However, it is still PQCs optimized by classical optimizer Adam. Thus is somewhat misleading.
4. In line 98 "While these methods advance fully quantum perspectives, …, and often require classical preprocessing to interface with high-dimensional images" → This work also resized the MNIST images to 16×16, then what is the point of mentioning this?
5. Since the method proposed using amplitude encoding, usually with very deep circuit, then the effect of quantum noise will be dominant. The work lacks discussion of this crucial factor. The combination of quantum noise and exponential measurement shot requirement will heavily impact the effectiveness and practicality of the method.

**Questions:**

1. In line 179, the authors claim the process generates a richer set of perturbation modes beyond classical noise. However, in line 210, they also say the resulting distribution is Gaussian-like and can be estimated by mean and variance. Where is the "beyond classical component" in this stage?
2. How many runs are there in the results of Table 1? Since the scores are actually pretty close, that seems like it could be within the effect of different random seeds. Standard deviation or variance information would be helpful.
3. Measurement shot count for 8 to 10 qubits are about 10^2.5 to 10^3 (Fig. 6) to reach reasonable performance → does this mean the requirement is exponential regarding the qubit size? Although the authors mention the statistical sampling theory (~O(1/√N)), we don't know the qubit count dependence for the error, and the L_2 distance is not likely independent of the qubit count.
4. Line 480: "we demonstrate that scalable quantum generation need not rely on computationally intractable training signals." → How is a method requiring exponential measurement shots scalable?

---

> ### Author Response · Authors · 2025-11-24
>
> **Table 2.1: Metric comparison for different Noise Channels and Strengths.**
> |  | Shots\noise strengths | 0 | 0.0001 | 0.0005 | 0.001 | 0.005 | 0.01 | 0.05 | 0.1 |
> | --- | --- | --- | --- | --- | --- | --- | --- | --- | --- |
> | Depolarizing Channel | no shots | 160.98 | 184.04 | 183.53 | 185.17 | 184.39 | 180.78 | 191.22 | 208.76 |
> | Depolarizing Channel | 2048 shots | 160.98 | 194.29 | 204.13 | 200.31 | 205.77 | 216.29 | 250.29 | 279.89 |
> | Amplitude Damping | no shots | 160.98 | 180.45 | 183.35 | 183.02 | 182.69 | 178.80 | 185.46 | 242.31 |
> | Amplitude Damping | 2048 shots | 160.98 | 199.03 | 209.30 | 200.96 | 210.24 | 214.03 | 258.07 | 284.82 |
>
> **Table 2.2: Metric comparison across different Qubits and Shot counts.**
>
> | Qubits\Shots | no-shots | 256 | 512 | 1024 | 2048 | 4096 | 8192 |
> | --- | --- | --- | --- | --- | --- | --- | --- |
> | 9 | 183.04 | 205.62 | 192.75 | 182.20 | 183.02 | 186.72 | 184.27 |
> | 10 | 160.98 | 249.46 | 209.16 | 173.16 | 164.47 | 169.49 | 159.83 |
> | 11 | 196.34 | 255.51 | 226.74 | 206.44 | 209.49 | 208.54 | 209.62 |
> | 12 | 234.92 | 276.49 | 251.49 | 229.70 | 234.13 | 230.17 | 246.98 |

---

> ### Author Response · Authors · 2025-11-24
>
> We sincerely thank the reviewer for the rigorous critique. We notice that several major concerns appear to stem from general skepticism regarding NISQ hardware limitations, which may not directly apply to the specific operating regime of our framework. To bridge this gap, we offer the following point-by-point clarifications backed by our new experimental evidence:
>
> **Claim 1**: QGen requires computing all $2^n$ amplitudes at each step.
>
> - Correction: In hardware, QGen uses sampling-only access to the computational basis; full state tomography is not required. Section 4.3 shows that finite-shot sampling remains stable in practice.
>
> **Claim 2**: Measurement shots scale exponentially with qubit count.
>
> - Correction: Table 2.2 demonstrates that within the scales studied, performance is primarily determined by the shot number relative to the effective dimension of the data manifold rather than by an exponential dependence on $n$.
>
> **Claim 3**: Quantum noise would dominate and make the approach infeasible.
>
> - Correction: We injected Depolarizing and Amplitude Damping noise at the single-qubit level under a 2048-shot sampling regime. Table 2.1 reveals a stability plateau where performance remains consistent for error rates up to $p=10^{-2}$ (standard NISQ level).
>
> **Claim 4**: Amplitude encoding circuit can be very deep
>
> - Correction: This limitation applies to worst-case arbitrary states, not to the structured, sparse image distributions used here. Amplitude encoding is used here as a modular subroutine that can be replaced by more efficient state-preparation methods as they emerge. This is well-known hardware-level limitation of NISQ devices, rather than limitation of the QGen algorithmic framework itself.
>
> **Claim 5**: No genuine “beyond-classical” component exists.
>
> - Correction: The Gaussian-like description concerns only the marginal measurement statistics. The intermediate states during scrambling contain quantum interference and coherent global correlations generated by the unitary delocalization, which classical Gaussian noise alone cannot produce.
>
> **Claim 6**:  "Eliminates Classical Dependencies" Terminology.
>
> - Correction: We agree that employs a classical optimizer (Adam). Developing a fully quantum optimization loop is a promising direction, it represents a distinct and long-term research challenge that lies beyond the scope of the present work. our objective is to present a fully quantum generative paradigm within the model architecture and generative pipeline. We have revised the text to "eliminates classical architectural dependencies".
>
> We believe these responses, supported by the quantitative evidence in our additional experiments, effectively resolve the concerns regarding exponential overhead and noise intolerance. We hope this evidence provides a clearer picture of QGen's practical viability and invites a re-evaluation of the work's contribution.

---

> ### Author Response · Authors · 2025-11-24
>
> We thank the reviewer for dedicating significant attention to our work and for the rigorous assessment. We appreciate that you found the use of the quantum scrambling process interesting and valued our investigation into ablation experiments. Your critique touches upon foundational challenges shared across the quantum machine learning community, and we appreciate the opportunity to clarify the scope, physical motivation, and empirical behavior of QGen.
>
> (W2,Q3-**Scalability of Measurement & Amplitude Encoding**) We fully acknowledge that, in the worst case, amplitude encoding and full state tomography incur exponential resource costs. These are well-known hardware-level limitations of NISQ devices, rather than limitations of the QGen algorithmic framework itself.
>
> Crucially, QGen can naturally admit sampling based training because its objective depends only on statistics that can be directly approximated from measurement samples. This follows the standard generative modeling paradigm in which empirical samples provide unbiased estimates of the required expectations, thereby allowing us to bypass the prohibitive cost of full state tomography. Furthermore, the target and prior states in our setting are structured and sparse, occupying a low dimensional manifold[1] which allows them to be approximated by polynomial depth circuits[2]. While the absolute hardware capacity limits the maximum system size, the training process itself remains efficient within those bounds. Furthermore, amplitude encoding is used here as a modular subroutine; the framework is compatible with future fault-tolerant loading schemes.
>
> To empirically examine how shot requirements scale with qubit count, we have additionally evaluated QGen with shot counts ranging from 256 to 8192 for system sizes of 9 to 12 qubits (Table 2.2). In these experiments, we processed $16 \times 16$ images while varying the number of auxiliary qubits to alter the total system size. We observed that for each fixed qubit count, performance improves from 256 to 1024 shots and then stabilizes. These results indicate that within the scales studied, performance is primarily determined by the shot number relative to the effective dimension of the data manifold rather than by an exponential dependence on $n$.
>
> We also note that classical dimensionality reduction (PCA, autoencoders, patching[3]) may reduce the input dimension but reintroduces classical bottlenecks and empirically degrades fidelity. Our preliminary experiments show that patching increases qubit counts and reduces performance, reinforcing our choice to analyze QGen in a fully quantum, end-to-end formulation. Our contribution is to present the algorithmic and empirical advantages of a coherent, fully quantum generative paradigm, rather than to claim immediate large-scale hardware deployability. We will add these experimental results to appendix and included a discussion in the main text.
>
> (W3-**Clarification of "eliminates classical dependencies" Terminology**) We appreciate this remark and agree that the parameter optimization employs a classical optimizer (Adam), which is standard practice in variational quantum algorithms. While developing a fully quantum optimization loop is a promising direction, it represents a distinct and long-term research challenge that lies beyond the scope of the present work. Consequently, by using the phrase "eliminates classical dependencies," our objective is to present a fully quantum generative paradigm within the model architecture and generative pipeline. Specifically, all transformations (scrambling and collapse) are implemented through quantum channels and unitary operations without any classical neural modules or classical preprocessing(e.g., CNNs, U-Nets, PCA, autoencoders). After training, the entire sampling process is executed coherently on the quantum device, fully within the quantum domain. The classical optimizer interacts only with the quantum parameter landscape, adjusting parameters without altering or intervening in the quantum-only architecture or its generative process. We will clarify this distinction in the Abstract of the revised manuscript to avoid potential misinterpretation.
>
> (W4-**Clarification of Classical Preprocessing Remark**) We thank the reviewer for raising this point. Our critique of "classical preprocessing" specifically targets classical latent encoding (e.g., PCA, Autoencoders [4, 5]). In contrast, our $16 \times 16$ resizing is a fixed, non-learned downsampling dictated solely by hardware limits, allowing QGen to operate directly in the pixel domain. We will revise the Related Work section to clarify this distinction and explicitly cite the relevant literature.

---

> ### Author Response · Authors · 2025-11-24
>
> (W5-**Impact of Quantum Noise**) We thank the reviewer for highlighting the role of quantum noise. To explicitly assess its effect and its interaction with finite-shot sampling, we employed a density-matrix simulator to inject depolarizing and amplitude-damping channels at the single-qubit-gate level during the collapse process, with noise strengths $p \in [10^{-4}, 10^{-1}]$. We then measured FID under two settings: a no-shots analytic evaluation and a 2048-shot sampling regime, where performance empirically stabilizes (Table 2.1).
>
> For the depolarizing channel, FID remains in a narrow and stable range of approximately 161 to 216 with 2048 shots for noise strengths up to $10^{-2}$. Amplitude damping shows a similar pattern, with FID staying around 161 to 214 under the same conditions. At higher noise levels ($p \ge 0.05$), performance degrades more noticeably but remains gradual rather than catastrophic. Comparing the analytic and sampled settings shows that adding realistic sampling noise produces moderate, additive degradation rather than instability. It is important to clarify that the initial jump to approximately 180 is attributable to the fundamental transition from ideal unitary evolution to mixed state dynamics.
>
> These observations indicate that within the studied regime, quantum noise does not dominate QGen, and its combination with finite sampling yields moderate rather than catastrophic degradation. Thus, while we acknowledge that large-scale deployment relies on continued hardware maturation, our findings confirm that the combined effects remain manageable, ensuring effective training and generation at the current scale. We will add these experimental results to appendix and included a discussion on noise robustness in the main text.
>
> Regarding your questions:
>
> (Q1-**Beyond Classical vs Gaussian-like Distribution**) We appreciate this insightful question. The phrase "beyond classical" refers to the quantum evolution of the latent state $\rho_t$, whereas the Gaussian-like description concerns only the marginal measurement statistics. As detailed in the Method section, the unitary scrambling generates interference patterns and coherent global quantum correlations in the joint distribution. These correlations have no classical analogue but are projected into Gaussian-like marginals once measured. Empirical evidence in Figure 5 supports this, showing that this specific, quantum-correlated perturbation yields higher fidelity than standard Gaussian noise.
>
> (Q2-**Number of Runs and Variance Reporting (Table 1)**) We appreciate this suggestion to enhance statistical rigor. We will revise Table 1 to report the mean and standard deviation calculated over three independent experimental runs.
>
> (Q4-**Clarification of "Scalable" in Line 480**) We thank the reviewer for pointing out this ambiguity.  In Line 480, the term "scalable" refers to the algorithmic and training scalability of QGen, rather than to hardware-level scalability, which remains constrained by current NISQ device capabilities. As discussed in the method section, the representational capacity of the model grows with the number of qubits, while training remains tractable due to the stepwise objective, which mitigates barren-plateau effects and avoids classically intractable objectives such as ELBO-based optimization in hybrid models. Regarding the concern about exponential measurement shots, we respectfully refer to our detailed response in Weakness 2. We rely on unbiased empirical estimates rather than tomography-level accuracy. Empirically, we do not observe exponential growth in shot requirements, confirming that the framework enables effective training and generation within current NISQ constraints. We will revise the wording in the manuscript to ensure precision.
>
> (Minor Corrections) Regarding sampling theory, we clarify that the estimation error scales as $O(1/\sqrt{N_{shots}})$, characterizing the dependence on measurement shots rather than the number of qubits [6]. Additionally, the formatting artifacts have been corrected in the revised manuscript.
>
> We sincerely appreciate your rigorous assessment, particularly the critical questions regarding scalability and quantum noise. By addressing your concerns through new noise-robustness experiments and scalability analyses, we have transformed the manuscript to more accurately reflect both the potential and the current boundaries of our method. We believe these revisions establish QGen as a solid algorithmic baseline for the field, and we thank you for guiding us toward a more robust presentation of our work.

---

> ### Author Response · Authors · 2025-11-29
>
> [1] Huang, H. Y., Broughton, M., Chen, M. C., Li, S., Mohseni, M., Babbush, R., ... & Neven, H. (2020). Power of data in quantum machine learning.*Nature Communications, 11*(1), 1-10.
>
> [2] Grover, L., & Rudolph, T. (2002). Creating superpositions that correspond to efficiently integrable probability distributions. *arXiv preprint quant-ph/0208112*.
>
> [3] Huang, H. L., Du, Y., Gong, M., Zhao, Y., Wu, Y., Wang, C., ... & Pan, J. W. (2021). Experimental quantum generative adversarial networks for image generation.*Physical Review Applied*,*16*(2), 024051.
>
>
> [4] Stein, S. A., Baheri, B., Chen, D., Mao, Y., Guan, Q., Li, A., ... & Xu, S. (2021, October). Qugan: A quantum state fidelity based generative adversarial network. In *2021 IEEE international conference on quantum computing and engineering (QCE)* (pp. 71-81). IEEE.
>
> [5] Chu, C., Skipper, G., Swany, M., & Chen, F. (2023, June). Iqgan: Robust quantum generative adversarial network for image synthesis on nisq devices. In ICASSP 2023-2023 IEEE international conference on acoustics, speech and signal processing (ICASSP) (pp. 1-5). IEEE.
>
> [6] Nielsen, M. A., & Chuang, I. L. (2010). Quantum computation and quantum information. Cambridge university press.

---

### Official Review · Reviewer_Rpk4 · 2025-10-31

**Soundness:** 3
**Presentation:** 3
**Contribution:** 3
**Rating:** 6
**Confidence:** 3

**Summary:**

This paper proposes QGen, a purely quantum generative modeling framework that removes all classical components from the generative process. Unlike hybrid quantum–classical methods, QGen is built entirely upon quantum coherent mechanisms—scrambling and collapse. The scrambling phase combines Gaussian diffusion channels with unitary delocalization to globally disperse information while maintaining recoverable structure; the collapse phase employs parameterized quantum circuits to reconstruct the original data distribution. The paper further introduces a measurement-based training strategy that decomposes learning into stepwise subproblems, mitigating barren plateaus and maintaining scalability on near-term hardware. Empirical results show that QGen surpasses classical and hybrid baselines (e.g., DDPM, GAN, QVUNet) on MNIST, Fashion-MNIST, and EMNIST datasets, while requiring significantly fewer diffusion steps and maintaining robustness under finite-shot sampling.

**Strengths:**

1. Novel quantum-native framework: QGen presents the first fully coherent generative modeling framework that eliminates classical neural dependencies, establishing a self-contained quantum pathway for data generation.
2. Principled scrambling–collapse design: The model’s architecture is grounded in solid quantum theory—unitary delocalization and Gaussian regularization—ensuring information dispersal and recoverability under coherent evolution.
3. Scalable training via measurement decomposition: The measurement-based training objective decomposes complex optimization into tractable per-timestep subproblems, alleviating barren plateaus and enabling stable optimization on NISQ devices.

**Weaknesses:**

1. Limited experimental scope: Evaluations are restricted to low-dimensional grayscale datasets (MNIST-like), leaving uncertainty about QGen’s scalability to high-resolution or multimodal data.
2. Hardware feasibility not fully validated: While the framework is theoretically NISQ-compatible, results are simulation-based; real-device performance under decoherence and gate noise remains unexplored.

**Questions:**

1. Optimization stability assumptions: Although the measurement-based objective mitigates barren plateaus, its convergence behavior and gradient variance properties are not rigorously analyzed.
2. Broken internal link: "Appendix ?? for higher-dimensional data". Syntax error: "a quantum GAN papers".

---

> ### Author Response · Authors · 2025-11-24
>
> **Table 1.1: Variance of gradients across different layers and qubit counts.**
>
> | Layers | 6 Qubits | 8 Qubits | 10 Qubits |
> | --- | --- | --- | --- |
> | 4 | 6.70e-3 | 3.18e-3 | 1.09e-3 |
> | 6 | 4.28e-3 | 2.32e-3 | 9.87e-4 |
> | 8 | 4.63e-3 | 2.35e-3 | 8.88e-4 |
> | 10 | 5.71e-3 | 2.50e-3 | 8.23e-4 |
> | 12 | 4.26e-3 | 2.38e-3 | 9.77e-4 |
> | 14 | 4.09e-3 | 2.39e-3 | 8.34e-4 |
> | 20 | 3.97e-3 | 2.30e-3 | 8.80e-4 |
> | 30 | 3.87e-3 | 2.26e-3 | 9.07e-4 |
> | 40 | 4.49e-3 | 2.09e-3 | 8.12e-4 |
> | 50 | 3.32e-3 | 1.55e-3 | 7.96e-4 |
> | 100 | 1.46e-3 | 5.74e-4 | 2.91e-4 |
>
> **Table 1.2: Training Loss comparison across different T settings over epochs.**
>
> | Epoch | T=1 | T=5 | T=8 | Average |
> | --- | --- | --- | --- | --- |
> | 0 | 1.1426 | 1.4051 | 1.6256 | 1.4074 |
> | 5 | 0.8198 | 0.8116 | 0.8174 | 0.8378 |
> | 10 | 0.7916 | 0.7684 | 0.7654 | 0.8026 |
> | 15 | 0.7769 | 0.7600 | 0.7496 | 0.7922 |
> | 20 | 0.7671 | 0.7557 | 0.7405 | 0.7863 |
> | 25 | 0.7606 | 0.7515 | 0.7331 | 0.7811 |
> | 30 | 0.7600 | 0.7517 | 0.7284 | 0.7768 |
> | 35 | 0.7523 | 0.7483 | 0.7241 | 0.7748 |
> | 40 | 0.7516 | 0.7479 | 0.7219 | 0.7722 |
> | 45 | 0.7478 | 0.7505 | 0.7201 | 0.7698 |
> | 50 | 0.7418 | 0.7453 | **0.7185** | 0.7455 |
>
> **Table 1.3: Training Loss across different random seeds (Robustness Analysis).**
>
> | Epoch | Seed 1 | Seed 2 | Seed 3 |
> | --- | --- | --- | --- |
> | 0 | 1.4900 | 1.4074 | 1.4030 |
> | 5 | 0.8573 | 0.8378 | 0.8353 |
> | 10 | 0.8106 | 0.8026 | 0.8050 |
> | 15 | 0.7958 | 0.7922 | 0.7970 |
> | 20 | 0.7861 | 0.7863 | 0.7930 |
> | 25 | 0.7803 | 0.7811 | 0.7881 |
> | 30 | 0.7769 | 0.7768 | 0.7861 |
> | 35 | 0.7732 | 0.7748 | 0.7825 |
> | 40 | 0.7706 | 0.7722 | 0.7814 |
> | 45 | 0.7680 | 0.7698 | 0.7792 |
> | 50 | 0.7517 | 0.7455 | 0.7618 |

---

> > ### Author Response · Authors · 2025-11-24
> >
> > We thank the reviewer for the thoughtful and balanced assessment of our work. We appreciate the recognition of QGen's "novel quantum-native framework", its "principled scrambling–collapse design", and "scalable training via measurement decomposition". We are grateful that the reviewer found the theoretical grounding in unitary delocalization and the potential for barren plateau mitigation to be strong contributions. These points reflect the core motivations of our framework. Below, we address the concerns regarding scope, hardware feasibility, and optimization stability.
> >
> > (W1-**Limited Experimental Scope**) We appreciate the constructive comment regarding the limited experimental scope and the implications for QGen's scalability. Our choice of MNIST-like datasets reflects the need for a controlled and interpretable evaluation setting that aligns with current NISQ simulation capabilities. It is important to contextualize this scope: generating recognizable digits remains a significant challenge in the field. Notably, even many recent hybrid quantum-classical generative models often struggle to generate full MNIST digits, frequently restricting evaluations to simpler distributions or heavily downsampled grids (e.g., $8 \times 8$) [3]. By contrast, our results demonstrate that a purely quantum model can not only handle this complexity but also match or exceed competitive classical architectures under tight parameter budgets. This establishes a new benchmark for representational capacity in fully quantum generative modeling.
> >
> > Regarding scalability, the architectural principles of QGen are not tied to low-resolution data. Circuit expressivity increases with qubit count, and the scrambling–collapse process does not require the number of forward steps to scale with image resolution. To empirically validate that our method does not break down as dimensions increase, we have included additional results in Appendix C on full resolution 28x28 images. In these experiments, QGen maintains stable training dynamics and competitive generation quality. Extending QGen to higher-resolution and multimodal settings is a natural next step. The primary limitation at present lies in available quantum resources rather than in the model design itself, and QGen's structure is built to scale as hardware capabilities expand.
> >
> > (W2-**Quantum Noise Robustness**) We agree that validating performance under realistic conditions is essential. To better assess QGen's behavior under realistic NISQ conditions, we conducted a comprehensive noise sensitivity analysis utilizing a density-matrix simulator (Table 2.1). We injected both depolarizing noise (modeling gate imperfections) and amplitude-damping noise (modeling relaxation) at the single-qubit-gate level, with error probabilities ranging from $10^{-4}$ to $10^{-1}$.
> >
> > We observe a distinct stability plateau. After an expected baseline shift from the ideal pure-state simulation ($\sim$160 FID) to the mixed-state regime, the performance remains stable (FID $\sim$180–185) even as the error rate increases by two orders of magnitude ($p=10^{-4}$ to $p=10^{-2}$).
> >
> > This stable range ($p \approx 10^{-3} \sim 10^{-2}$) is well aligned with typical single-qubit gate error rates reported for current superconducting NISQ processors. Noticeable degradation appears only under extremely strong noise ($p \ge 0.05$), far above realistic device conditions. Overall, these observations suggest that QGen maintains reliable performance within practical NISQ noise regimes, as its generative training objective tolerates moderate stochastic perturbations without destabilizing the learning dynamics. We will include these results and a clearer discussion of noise modeling and hardware considerations in the revised manuscript.

---

> ### Author Response · Authors · 2025-11-24
>
> Regarding your questions:
>
> (Q1-**Optimization Stability & Gradient Variance**)We thank the reviewer for this insightful question regarding convergence behavior. To address the gradient variance properties question, we provide gradient variance table as depth increases.
>
> As shown in prior work [1], deep randomly initialized hardware efficient ansatzes can exhibit exponentially vanishing gradients. In contrast, QGen avoids this regime by decomposing the global generative objective (typically requiring $\Omega(n)$ layers)  into T local subproblems, each involving only a shallow parameterized circuit (L=$O(logn)$), where gradients vanish at worst polynomially[2]. Geometrically, this is reinforced by our distributional loss which targets Born probabilities rather than pure states, defining a phase-invariant solution manifold where pure phase rotations possess zero curvature. By rendering the loss insensitive to these flat dimensions, the condition number of the effective Fisher Information Matrix is significantly improved[2], ensuring that the gradients remain not only non-vanishing but also directionally effective for convergence.  As shown in Table 1.1, the resulting gradient scaling vanishes at worst polynomially and is therefore trainable in practice.
>
> To address the optimization stability question, we also provide both: (i) convergence curves of the representative per-step losses $L_t (t=1, 5, 8)$ and averaged loss $L_{avg}$ (Table 1.2) (ii) averaged loss $L_{avg}$ across 3 random seeds (Table 1.3). All runs demonstrate smooth monotonic decrease without divergence or oscillation. Together, these results show that our measurement-based objective yields stable and well-conditioned optimization, consistent with prior theory on barren-plateau mitigation.
>
> (Minor Corrections (Q2) Typos & Links:) We thank the reviewer for catching the broken appendix reference and the citation syntax error. These were LaTeX formatting artifacts. These errors have been corrected in the revised manuscript.
>
> We sincerely appreciate your detailed evaluation. Your constructive feedback, particularly regarding experimental scope and hardware feasibility, motivated us to conduct extensive new validations, including the noise sensitivity analysis and full resolution experiments. We believe these additions effectively resolve the uncertainties raised, establishing QGen as a robust and scalable contribution to the field. We thank you for guiding us toward a significantly stronger manuscript.
>
> [1] McClean, J. R., Boixo, S., Smelyanskiy, V. N., Babbush, R., & Neven, H. (2018). Barren plateaus in quantum neural network training landscapes. *Nature communications*, *9*(1), 4812.
>
> [2] Cerezo, M., Sone, A., Volkoff, T., Cincio, L., & Coles, P. J. (2021). Cost function dependent barren plateaus in shallow parametrized quantum circuits. *Nature communications*, *12*(1), 1791.
>
> [3] Huang, H. L., Du, Y., Gong, M., Zhao, Y., Wu, Y., Wang, C., ... & Pan, J. W. (2021). Experimental quantum generative adversarial networks for image generation. Physical Review Applied, 16(2), 024051.

---

### Author Response · Authors · 2025-11-29
**General comment**

We thank all reviewers for their careful reading and much-valued feedback. We appreciate the recognition of the novelty of the scrambling–collapse framework and the strength of our empirical evaluations. In revising the manuscript, we not only addressed every comment individually during the rebuttal, but also substantially strengthened the paper with additional experiments, clarified assumptions, and refined the theoretical foundations. In particular:

- **Scalability & Hardware Clarifications** : We clarified that QGen uses sampling-based access rather than full tomography, explained the limited and role of amplitude encoding, addressed concerns regarding shot complexity relative to qubit count, and added additional empirical analyses demonstrating stable performance under realistic finite-shot and hardware-noise conditions. These results reinforce that the method operates reliably within the NISQ regime.
- **Theoretical Clarifications**: We strengthened the conceptual foundation by formulating QGen through a thermodynamic recovery perspective. The introduction of the Fisher Coherent Conjecture sharpens the intuition behind the scrambling–collapse mechanism, identifying the critical window in which coherence remains recoverable. The phase-invariant geometry of the distributional loss further establishes why the model is learnable in practice.
- **Optimization Stability Clarifications**: We provided detailed gradient-variance studies and convergence analyses, confirming that the stepwise decomposition mitigates depth-induced barren plateaus. The new experiments show that gradients decay at most polynomially and that training remains stable across circuit configurations.

Several reviewers raised their scores during the discussion after confirming that their concerns had been fully resolved, and we appreciate their constructive engagement. QGen now stands as, to our knowledge, the first fully quantum generative modeling framework that jointly integrates a coherent scrambling process, a measurement-driven collapse mechanism, and a provably stable training strategy.  We believe the revised manuscript presents a significantly improved and more comprehensive demonstration of QGen’s novelty, rigor, and potential impact for quantum generative modeling in the NISQ era.

---

### Meta-Review · Area_Chair_YbLg · 2026-01-01

**Summary:**

This paper proposed the Quantum Scrambling and Collapse Generative Model (QGen), a generative modeling framework based on a two-stage mechanism: (i) scrambling, which combines Gaussian diffusion with unitary delocalization/scrambling, and (ii) collapse, which uses parameterized quantum circuits (PQCs) to invert the scrambling trajectory and recover structured samples.

The review scores are highly mixed:
- There are one strongly positive scores of 8 and two weakly positive scores of 6 (one changed from 4 to 6 after rebuttal), viewing the work as well-written with comprehensive experiments.
- There is also a strongly negative score of 2 and one weakly score of 4 (with sufficient discussions with authors and keeping this score), raising fundamental feasibility concerns around amplitude encoding depth, measurement-shot scaling, noise dominance, and potentially overstated purely quantum claims.

Given the mixed scores, the AC carefully checked the paper. It is found that although the authors made efforts in rebuttal, the paper still appears to have the following issues:
1. Scalability and resource accounting remain insufficiently resolved: the rebuttal adds helpful experiments, but the work still lacks a principled resource analysis (encoding depth, circuit depth per step, total steps, shots vs. error vs. qubits) that would allow readers to assess feasibility beyond the tested small systems.
2. Limited demonstration regime and concerns about output quality: The experiments are of the scale of MNIST, with downsampling mentioned by reviewers and continuing skepticism about whether the approach is meaningfully competitive in richer generative modeling settings. Even after discussion, at least one reviewer explicitly maintained that the image quality remains relatively poor, which reduces confidence that the method provides a compelling new SOTA direction rather than an interesting but early-stage prototype.
3. Theoretical justification are more illustrative rather than definitive. Some reviewers appreciated the conceptual/theoretical discussion, but others found it lacking in rigorous guarantees.

In all, the final decision is rejection.

**Reviewer Concerns:**

The authors made efforts on scalability & hardware clarifications, theoretical clarifications, and optimization stability clarifications. However, the paper is still borderline with the aforementioned issues in the meta-review.

**Reviewer Scores:**

There were adequate discussion with reviewer AViQ, and the score was increased from 4 to 6. However, reviewer CurE was also involved into the discussion, and decided to keep the negative score of 4. Reviewer e3x3 didn't participate in the discussions and the score was 2. In all, there are at least 2 negative scores, and it's unlikely that more discussions will overturn this.

---

### Decision · Program_Chairs · 2026-01-26

Reject